# Evaluation of a Novel Boron-Containing α-d-Mannopyranoside for BNCT

**DOI:** 10.3390/cells9051277

**Published:** 2020-05-21

**Authors:** Takao Tsurubuchi, Makoto Shirakawa, Wataru Kurosawa, Kayo Matsumoto, Risa Ubagai, Hiroshi Umishio, Yasuyo Suga, Junko Yamazaki, Akihiro Arakawa, Yutaka Maruyama, Takuya Seki, Yusuke Shibui, Fumiyo Yoshida, Alexander Zaboronok, Minoru Suzuki, Yoshinori Sakurai, Hiroki Tanaka, Kei Nakai, Eiichi Ishikawa, Akira Matsumura

**Affiliations:** 1Department of Neurosurgery, Faculty of Medicine, University of Tsukuba, 1-1-1 Tennodai, Tsukuba 305-8575, Japan; f40072@fukuyama-u.ac.jp (M.S.); yoshida.fumiyo.ff@u.tsukuba.ac.jp (F.Y.); knakai@pmrc.tsukuba.ac.jp (K.N.); e-ishikawa@md.tsukuba.ac.jp (E.I.); matsumura.akira.ft@alumni.tsukuba.ac.jp (A.M.); 2Department of Pharmaceutical Sciences, University of Fukuyama, 1 Sanzo, Gakuen-cho, Fukuyama 729-0292, Japan; 3Institute for Innovation, Ajinomoto Co., Inc., 1-1 Suzukichō, Kawasaki-ku, Kawasaki 210-8681, Japan; wataru_kurosawa@ajinomoto.com (W.K.); kayo_matsumoto@ajinomoto.com (K.M.); risa_ubagai@ajinomoto.com (R.U.); hiroshi_umishio@ajinomoto.com (H.U.); yasuyo_suga@ajinomoto.com (Y.S.); junko_yamazaki@ajinomoto.com (J.Y.); akihiro_arakawa@ajinomoto.com (A.A.); yutaka_maruyama@ajinomoto.com (Y.M.); takuya_seki@ajinomoto.com (T.S.); yusuke_shibui@ajinomoto.com (Y.S.); 4Institute for Integrated Radiation and Nuclear Science, Kyoto University, 2 Asashiro-Nishi, Kumatori-cho, Sennan-gun, Osaka 590-0494, Japan; msuzuki@rri.kyoto-u.ac.jp (M.S.); yosakura@rri.kyoto-u.ac.jp (Y.S.); h-tanaka@rri.kyoto-u.ac.jp (H.T.); 5Department of Radiation Oncology, Faculty of Medicine, University of Tsukuba, 1-1-1 Tennodai, Tsukuba 305-8575, Japan

**Keywords:** boron-containing α-d-mannopyranoside, closo-dodecarborate, low-molecular-weight compound, BNCT

## Abstract

Boron neutron capture therapy (BNCT) is a unique anticancer technology that has demonstrated its efficacy in numerous phase I/II clinical trials with boronophenylalanine (BPA) and sodium borocaptate (BSH) used as ^10^B delivery agents. However, continuous drug administration at high concentrations is needed to maintain sufficient ^10^B concentration within tumors. To address the issue of ^10^B accumulation and retention in tumor tissue, we developed MMT1242, a novel boron-containing α-d-mannopyranoside. We evaluated the uptake, intracellular distribution, and retention of MMT1242 in cultured cells and analyzed biodistribution, tumor-to-normal tissue ratio and toxicity in vivo. Fluorescence imaging using nitrobenzoxadiazole (NBD)-labeled MMT1242 and inductively coupled mass spectrometry (ICP-MS) were performed. The effectiveness of BNCT using MMT1242 was assessed in animal irradiation studies at the Kyoto University Research Reactor. MMT1242 showed a high uptake and broad intracellular distribution in vitro, longer tumor retention compared to BSH and BPA, and adequate tumor-to-normal tissue accumulation ratio and low toxicity in vivo. A neutron irradiation study with MMT1242 in a subcutaneous murine tumor model revealed a significant tumor inhibiting effect if injected 24 h before irradiation. We therefore report that ^10^B-MMT1242 is a candidate for further clinical BNCT studies.

## 1. Introduction

Boron neutron capture therapy (BNCT) is a binary anticancer technology that was first proposed by Locher in 1936 and has since been further developed and improved by a number of research groups all over the world. It commands attention as a state-of-the-art, minimally invasive cancer treatment that functions as a single cell level-selective tumoricidal therapy [1,2,3,4,5,6,7]. It employs a stable boron-10 (^10^B) isotope-containing compound injected into the patient and neutron irradiation produced at a nuclear reactor (or accelerator) and once the appropriate form of ^10^B-boron-containing compound has selectively accumulated in malignant tumor cells by a drug delivery system (DDS), the subsequent neutron irradiation of the tumor area induces alpha particles and Li nuclei emission within a 10 micrometer span, limiting the BNCT effect to tumor cells with accumulated boron and sparing the surrounding normal cells without boron accumulation [2,3,4,8,9]. This tumor-selective, high-dose irradiation may improve local control and survival after the initial treatment. Phase I/II clinical trials of BNCT were conducted for patients with glioblastoma and the early results suggest that BNCT is effective in selected cases [10,11,12].

Until recently, clinical studies using boronophenylalanine (BPA) and sodium borocaptate (BSH) as ^10^B delivery agents for the neutron capture reaction have been conducted [3,8,10,13]. However, continuous administration at high concentrations is needed to maintain sufficient ^10^B concentration within tumors [14]. For effective BNCT, ≥ 20 μg of ^10^B per gram of tumor must be selectively delivered to the tumor cells (~10^9^ atoms per cell), reside intracellularly for a sufficient amount of time, and simultaneously clear from surrounding normal tissues to ideally attain a tumor-to-normal tissue ratio of 3,4 to 1 [2,3,4,5,15,16]. To fulfill these criteria, various functionalized carborane derivatives have been recently developed, based on vision of carboranes as molecular moieties of choice for the synthesis of boron-delivery agents for BNCT due to their catabolic stability, relatively low toxicity, and a high boron content compared to BPA [17,18].

We have developed novel boron-containing low-molecular-weight compounds efficient for accumulation and retention in tumors, one of which is a boron-containing α-d-mannopyranoside derivative with three *closo*-dodecarborates, each carrying 12 boron atoms. This mannopyranoside compound carries the capability of a more active uptake by malignant tumor cells due to the Warburg effect and, coupled with several previous reports related to mannopyranoside [19,20,21,22,23], suggests a potential usefulness for BNCT. Additionally, overexpression of GLUT1 on glioma cells can give an edge for enhancing cargo delivery to glioma cells [24,25,26,27] as, among all other glucose analogs, mannose can be better recognized by these transporters [28]. It is transported into mammalian cells via facilitated diffusion effect by hexose transporters of the SLC2A group (GLUT) present primarily on the plasma membrane [22]. Mannose and its related compounds are also uptaken by mannose receptors broadly expressed in many types of cancer cells [29,30,31]. The Warburg effect, by which anaerobic glycolysis is upregulated in the growing tumor cells and tissues of malignant tumors, relies on glucose, fructose, galactose, and also mannose as the main nutritive supplies [32,33,34]. Recently, a saccharide-conjugated carborane compound (galactosyl carborane) [35], boronated carbohydrates [36,37,38,39], and carbohydrate derivatives of the closo-dodecaborate anion [40,41,42] were reported as specific targeting agents for potential application to BNCT.

Our aim is to develop low-molecular-weight, boron-containing carriers appropriate for BNCT that are easily and selectively uptaken by active transporters in malignant tumor cells while being well retained. To confirm the tumor selectivity and intratumoral tissue accumulation of this novel boron compound, in vitro and in vivo uptake studies were conducted together with biodistribution studies before confirming the effectiveness of this compound in BNCT using animal irradiation studies.

## 2. Materials and Methods

We used the boron compound MMT1242, which is an α-d-mannopyranoside derivative with three *N*,*N*-dimethylformamide (Figure 1a). MMT1242 containing ^10^B isotope was used in fluorescence imaging of ^10^B distribution in vivo and in irradiation experiments while, in all other experiments, MMT1242 containing natural boron was used. All in vivo experiments were approved by the Institutional Animal Care and Use Committee, the University of Tsukuba (Approval #18-127) and the Animal Experiment Institution Review Board of Ajinomoto Co., Inc. (Approval #2015106, 2017092, 2017153).

### 2.1. Chemical Synthesis

For chemical synthesis, cesium dodecaborate and ^10^B-enriched cesium dodecaborate were purchased from Katchem Ltd. (Prague, Czech Republic), QuadraPure^®^ IDA was obtained from Sigma-Aldrich, Inc. (St. Louis, MO, USA), Amberlite^®^ IR120B Na was obtained from Organo Corp. (Tokyo, Japan), all other reagents, including methyl α-d-mannopyranoside and 4-fluoro-7-nitro-2,1,3-benzoxadiazole, were purchased from Tokyo Chemical Industry Co., Ltd. (Tokyo, Japan), and all the solvents were purchased from Kanto Chemical Co., Inc. (Tokyo, Japan).

The synthesis of MMT1242 and NBD1242 are shown in Figure 1 and Figure 2. The *closo*-dodecaborate unit (5) that was previously synthesized by Orlova et al. (2007) and Semioshkin et al. (2007) was here prepared according to our methods described in the patent (WO2017164334) and, according to our modifications, contains a cesium salt to provide sufficient water solubility in contrast to the previously reported compound that contained a tetrabutylammonium salt [43,44]. ^1^H-NMR spectra were obtained on a Bruker Ascend™ 400 (Bruker Corp., Billerica, MA, USA). ESI-MS spectra were obtained using a Waters ZQ 2000 mass spectrometer equipped with an ESI ion source (Waters Corp., Milford, MA, USA). Preparative HPLC was performed on an InertSustain^®^ 5 µm C18 column (20 mm × 250 mm), using a GL science HPLC system PLC761 (GL Sciences Inc., Tokyo, Japan) and purification was accomplished using a water (20 mM Ammonium hydrogen carbonate)/acetonitrile gradient at 10 mL/min flow rate. Analytical HPLC for purity confirmation was carried out with an InertSustain^®^ 5 µm C18 column (4.6 mm × 250 mm, GL Sciences Inc.), using a Shimadzu Prominence HPLC system (Shimadzu Corp., Kyoto, Japan), in water (20 mM ammonium hydrogen carbonate)/acetonitrile gradient at 1 mL/min flow rate (95/5 to 40/60 over 30 min). The purities of MMT1242 and NBD1242 were >97%.

#### 2.1.1. Synthesis of MMT1242

##### Methyl 6-*O*-(tert-butyldiphenylsilanyl)-α-d-Mannopyranoside (**1**) [45,46]

To a solution of methyl α-d-mannopyranoside (5.00 g, 25.8 mmol) in *N*,*N*-dimethylformamide (5 mL) was added imidazole (2.11 g, 31.0 mmol) at room temperature. After *tert*-butylchlorodiphenylsilane (7.81 g, 28.4 mmol) was added at 0 °C (ice-water bath), the mixture was stirred at room temperature overnight. The reaction mixture was diluted with diethyl ether and washed with water, followed by 10% citric acid and 15% brine. Then, the organic layer was dried over anhydrous magnesium sulfate. After filtration, the filtrate was concentrated under reduced pressure. The obtained residue was purified by silica gel column chromatography using ethyl acetate/methanol to give compound **1** (12.5 g, 29.0 mmol, quantitative yield).

^1^H NMR (400 MHz, chloroform-*d*) δ 7.74–7.67 (m, 4H, phenyl -CH), 7.49–7.38 (m, 6H, phenyl -CH), 4.70 (d, *J* = 1.6 Hz, 1H, Man -CH), 4.00–3.89 (m, 3H, Man -CH), 3.85–3.78 (m, 2H, Man -CH_2_-), 3.64 (dt, *J* = 10.1, 5.3 Hz, 1H, Man -CH), 3.33 (s, 3H, -OCH_3_), 1.09 (s, 9H, -CH_3_).

##### Methyl 6-*O*-(tert-butyldiphenylsilanyl)-2,3,4-tris-*O*-(prop-2-ynyl)-α-d-Mannopyranoside (**2**)

To a mixture of sodium hydride (1.00 g, 60% suspension in oil, 25.0 mmol) and dehydrated *N*,*N*-dimethylformamide (1.5 mL), a solution of compound **1** (3.00 g, 6.94 mmol) in *N*,*N*-dimethylformamide (9 mL) was added at 0 °C (ice-water bath). After stirring of the mixture at room temperature for 20 min, propargyl bromide (2.35 mL, 31.2 mmol) and tetrabutylammonium iodide (933 mg, 2.5 mmol) were added at 0 °C (ice-water bath), and the mixture was stirred at room temperature overnight. The reaction mixture was diluted with diethyl ether and washed with water, followed by 10% citric acid and 15% brine. Then, the organic layer was dried over anhydrous magnesium sulfate. After filtration, the filtrate was concentrated under reduced pressure. The obtained residue was purified by silica gel column chromatography using hexane/ethyl acetate to give compound **2** (2.60 g, 4.75 mmol, 68%).

^1^H NMR (400 MHz, chloroform-*d*) δ 7.75–7.71 (m, 4H, phenyl -CH), 7.44–7.33 (m, 6H, phenyl -CH), 4.81 (d, *J* = 1.8 Hz, 1H, Man -CH), 4.46–4.24 (m, 6H, -OCH_2_CCH), 4.04 (dd, *J* = 3.2, 1.9 Hz, 1H, Man -CH), 4.00–3.89 (m, 3H, Man -CH, Man -CH_2_-), 3.75 (t, *J* = 9.5 Hz, 1H, Man -CH), 3.61 (ddd, *J* = 10.0, 5.2, 2.0 Hz, 1H, Man -CH), 3.36 (s, 3H, -OCH_3_), 2.47–2.44 (m, 2H, -CCH), 2.26 (t, *J* = 2.4 Hz, 1H, -CCH), 1.05 (s, 9H, -CH_3_).

##### Methyl 2,3,4-tris-*O*-(prop-2-ynyl)-α-d-Mannopyranoside (**3**)

To a solution of compound **2** (2.58 g, 4.72 mmol) in tetrahydrofuran (24 mL) was added 1M tetrabutylammonium fluoride in tetrahydrofuran (9.44 mL, 9.44 mmol) at 0 °C (ice-water bath). After stirring of the mixture at room temperature for 6 h, the reaction mixture was diluted with dichloromethane, and washed with water followed by 15% brine. Then, the organic layer was dried over anhydrous magnesium sulfate. After filtration, the filtrate was concentrated under reduced pressure. The obtained residue was purified by silica gel column chromatography using hexane/ethyl acetate to give compound **3** (1.32 g, 4.29 mmol, 91%).

^1^H NMR (400 MHz, chloroform-*d*) δ 4.81 (d, *J* = 1.9 Hz, 1H, Man -CH), 4.51–4.32 (m, 6H, -OCH_2_CCH), 4.07 (dd, *J* = 3.1, 1.9 Hz, 1H, Man -CH), 3.98–3.82 (m, 3H, Man -CH, Man -CH_2_-), 3.78 (t, *J* = 9.5 Hz, 1H, Man -CH), 3.58 (ddd, *J* = 9.6, 4.6, 3.1 Hz, 1H, Man -CH), 3.39 (s, 3H, -OCH_3_), 2.52–2.46 (m, 3H, -CCH), 2.06 (s, 1H, -OH).

##### Methyl 6-*O*-benzyl-2,3,4-tris-*O*-(prop-2-ynyl)-α-d-Mannopyranoside (**4**)

To a solution of compound **3** (150 mg, 0.486 mmol) in *N,N*-dimethylformamide (2.4 mL) was added sodium hydride (35.0 mg, 60% suspension in oil, 0.583 mmol) at 0 °C (ice-water bath), and the mixture was stirred at room temperature for 30 min. Then, benzyl bromide (0.087 mL, 0.73 mmol) was added at 0 °C (ice-water bath), and the mixture was stirred at room temperature for 3 h. After adding methanol (0.010 mL), the mixture was poured into water (20 mL) and extracted with ethyl acetate. The organic layer was washed with saturated brine and dried over anhydrous magnesium sulfate. After filtration, the filtrate was concentrated under reduced pressure. The obtained residue was purified by silica gel column chromatography using hexane/ethyl acetate to give compound **4** (102 mg, 0.255 mmol, 53%).

^1^H NMR (400 MHz, chloroform-*d*) δ 7.40–7.27 (m, 5H, phenyl -CH), 4.81 (d, *J* = 1.9 Hz, 1H, Man -CH), 4.65 (d, *J* = 12.1 Hz, 1H, -OCH_2_phenyl), 4.57 (d, *J* = 12.0 Hz, 1H, -OCH_2_phenyl), 4.47–4.21 (m, 6H, -OCH_2_CCH), 4.04 (dd, *J* = 3.3, 1.9 Hz, 1H, Man -CH), 3.92 (dd, *J* = 8.7, 3.3 Hz, 1H, Man -CH), 3.84–3.67 (m, 4H, Man -CH, Man -CH_2_-), 3.37 (s, 3H, -OCH_3_), 2.46 (t, *J* = 2.4 Hz, 1H, -CCH), 2.44 (t, *J* = 2.4 Hz, 1H, -CCH), 2.37 (t, *J* = 2.4 Hz, 1H, -CCH).

##### Methyl 6-*O*-benzyl-2,3,4-tris-*O*-[[1-[2-[2-(undecahydro-closo-dodecaboranyloxy)ethoxy]ethyl] triazol-4-yl]methyl]-α-d-Mannopyranoside Hexasodium Salt (MMT1242)

To a solution of compound **4** (102 mg, 0.255 mmol) in *tert*-butanol/water (1/2, *v*/*v*, 9.9 mL in total) was added *closo*-dodecaborate unit **5** (411 mg, 0.765 mmol), copper (II) sulfate pentahydrate (47.7 mg, 0.191 mmol), and sodium l-ascorbate (84.1 mg, 0.574 mmol). After stirring of the mixture at 70 °C overnight, the reaction mixture was cooled to room temperature. Then, water (10 mL) and a metal scavenger-supporting resin (QuadraPure^®^ IDA, 800 mg) were added to the mixture. After stirring of the mixture for 3 h, the resin was removed by filtration and the filtrate was concentrated under reduced pressure. The obtained residue was purified by preparative HPLC and those fractions containing the desired product were collected and lyophilized. The obtained residue was dissolved in water and treated with Amberlite^®^ IR120B Na. The obtained aqueous solution was lyophilized to give MMT1242 (85.9 mg, 0.0637 mmol, 25%).

MS (ESI, *m*/*z*): 651.0 [M + 6Na-2Na]^2−^; 1372 [M + 7Na]^+^. C_35_H_83_B_36_N_9_Na_6_O_12_. Calculated: *m*/*z* 651.5 [M + 6Na-2Na]^2−^; 1372 [M + 7Na]^+^.

^1^H NMR (400 MHz, deuterium oxide) δ 8.11 (s, 1H, triazole -CH), 8.07 (s, 1H, triazole -CH), 7.68 (s, 1H, triazole -CH), 7.33–7.24 (m, 5H, phenyl -CH), 4.79–4.73 (m, 4H), 4.63–4.56 (m, 1H, Man -CH), 4.56–4.49 (m, 5H), 4.44–4.30 (m, 4H), 3.94–3.73 (m, 8H), 3.68–3.38 (m, 17H, -OCH_2_-), 3.32 (s, 3H, -OCH_3_), 1.88–0.34 (m, 33H, -B_12_H_11_).

#### 2.1.2. Synthesis of NBD1242

##### 2-[[(3*R*,4*S*,6*S*)-6-methoxy-3,4,5-tris(prop-2-ynoxy)tetrahydropyran-2-yl]methyl]isoindoline-1,3-Dione (**6**)

To a solution of compound **3** (426 mg, 1.38 mmol) in tetrahydrofuran (14 mL) was added triphenylphosphine (543 mg, 2.07 mmol), diethyl azodicarboxylate (0.940 mL, 40% in toluene, 2.07 mmol), and phthalimide (305 mg, 2.07 mmol) at 0 °C (ice-water bath). After stirring of the mixture at room temperature for 40 min, triphenylphosphine (181 mg, 0.690 mmol), diethyl azodicarboxylate (0.627 mL, 40% in toluene, 1.38 mmol), and phthalimide (102 mg, 0.690 mmol) were added at 0 °C (ice-water bath). The mixture was stirred at room temperature for 25 min then concentrated under reduced pressure. The obtained residue was purified by silica gel column chromatography using hexane/ethyl acetate to give compound **6** (743 mg, 1.70 mmol) as crude material.

^1^H NMR (400 MHz, chloroform-*d*) δ 7.85 (dd, *J* = 5.4, 3.0 Hz, 2H, phthalimide -CH), 7.70 (dd, *J* = 5.4, 3.1 Hz, 2H, phthalimide -CH), 4.69 (d, *J* = 1.9 Hz, 1H, Man -CH), 4.57–4.22 (m, 6H, -OCH_2_CCH), 4.11 (dd, *J* = 13.6, 3.8 Hz, 1H, Man -CH), 4.05–3.90 (m, 4H, Man -CH_2_-, Man -CH), 3.70 (t, *J* = 9.2 Hz, 1H, Man -CH), 3.15 (s, 3H, -OCH_3_), 2.46 (t, *J* = 2.4 Hz, 1H, -CCH), 2.41 (t, *J* = 2.4 Hz, 1H, -CCH), 2.38 (t, *J* = 2.4 Hz, 1H, -CCH).

##### [(3*R*,4*S*,6*S*)-6-methoxy-3,4,5-tris(prop-2-ynoxy)tetrahydropyran-2-yl]methanamine trifluoroacetate (**7**)

To a solution of compound **6** (604 mg, 1.38 mmol) in ethanol (7 mL) was added hydrazine monohydrate (88.5 mg, 2.76 mmol). The mixture was stirred at room temperature overnight. After filtration, the filtrate was concentrated under reduced pressure. The obtained residue was purified by reverse-phase HPLC (water/acetonitrile, containing 0.05% trifluoroacetic acid) to give a mixture of compound **7**. The mixture was diluted with ethyl acetate and extracted with 0.2N HCl. After concentration of the aqueous layer under reduced pressure, the obtained residue was purified by reverse phase HPLC (water/acetonitrile, containing 0.05% trifluoroacetic acid) to give compound **7** (211 mg, 0.686 mmol, 50%).

^1^H NMR (400 MHz, methanol-*d*_4_) δ 4.88 (d, *J* = 2.0 Hz, 1H, Man -CH), 4.52–4.33 (m, 6H, -OCH_2_CCH), 4.08 (dd, *J* = 3.1, 1.9 Hz, 1H, Man -CH), 3.94 (dd, *J* = 8.9, 3.1 Hz, 1H, Man -CH), 3.77–3.62 (m, 2H, Man -CH_2_-), 3.47–3.41 (m, 4H, Man -CH, -OCH_3_), 3.11 (dd, *J* = 13.1, 8.7 Hz, 1H, Man -CH), 2.98–2.91 (m, 3H, -CCH).

##### N-[[(3*R*,4*S*,6*S*)-6-methoxy-3,4,5-tris(prop-2-ynoxy)tetrahydropyran-2-yl]methyl]-4-nitro-2,1,3-benzoxadiazol-7-amine (**8**)

To a solution of compound **7** (96.4 mg, 0.314 mmol) in *N*,*N*-dimethylformamide (1 mL) was added triethylamine (127 mg, 1.26 mmol) and 4-fluoro-7-nitro-2,1,3-benzoxadiazole (86.1 mg, 0.470 mmol). After stirring of the mixture at 70 °C overnight, the reaction mixture was cooled to room temperature, diluted with ethyl acetate, and then washed with water, followed by saturated brine. The organic layer was dried over anhydrous magnesium sulfate. After filtration, the filtrate was concentrated under reduced pressure. The obtained residue was purified by silica gel column chromatography using hexane/ethyl acetate to give compound **8** (56.3 mg, 0.120 mmol, 38%).

^1^H NMR (400 MHz, chloroform-*d*) δ 8.51 (d, *J* = 8.7 Hz, 1H, NBD -CH), 6.67 (s, 1H, -NH), 6.32 (d, *J* = 8.7 Hz, 1H, NBD -CH), 4.83 (d, *J* = 1.9 Hz, 1H, Man -CH), 4.55–4.30 (m, 6H, -OCH_2_CCH), 4.08 (dd, *J* = 3.1, 1.9 Hz, 1H, Man -CH), 4.01–3.88 (m, 2H, Man -CH_2_-), 3.84–3.75 (m, 3H, Man -CH), 3.35 (s, 3H, -OCH_3_), 2.50–2.47 (m, 2H, -CCH), 2.46 (t, *J* = 2.4 Hz, 1H, -CCH).

##### *N*-[[(3*R*,4*S*,6*S*)-6-methoxy-3,4,5-tris-*O*-[[1-[2-[2-(undecahydro-closo-dodecaboranyloxy)ethoxy]ethyl]triazol-4-yl]methyl]-tetrahydropyran-2-yl]methyl]-4-nitro-2,1,3-benzoxadiazol-7-amine hexasodium salt (NBD1242)

An operation similar to that of MMT1242 was performed using compound **8** (56.3 mg, 0.120 mmol) as a starting material to give NBD1242 (38.0 mg, 0.0273 mmol, 19%).

MS (ESI, *m*/*z*): 687.9 [M + 6Na-2Na]^2−^; 1397 [M + 6Na-Na]^−^. C_34_H_79_B_36_N_13_Na_5_O_14_. Calculated: *m*/*z* 687.5 [M + 6Na-2Na]^2−^; 1398 [M + 6Na-Na]^−^.

^1^H NMR (400 MHz, deuterium oxide) δ 8.48 (d, *J* = 9.0 Hz, 1H, NBD -CH), 8.17 (s, 1H, triazole -CH), 8.14 (s, 1H, triazole -CH), 8.05 (s, 1H, triazole -CH), 6.29 (d, *J* = 9.1 Hz, 1H, NBD -CH), 4.85–4.76 (m, 4H), 4.67–4.60 (m, 2H), 4.60–4.53 (m, 4H), 4.43–4.37 (m, 2H), 3.96 (t, *J* = 2.6 Hz, 1H, Man -CH), 3.91–3.73 (m, 8H), 3.72–3.65 (m, 1H, Man -CH), 3.65–3.30 (m, 15H, -OCH_2_-), 3.23 (s, 3H, -OCH_3_), 1.81–0.26 (m, 33H, -B_12_H_11_).

### 2.2. Intracellular Distribution of the MMT1242 In Vitro

To confirm the intracellular distribution of the compound in tumor cells, MMT1242 was labeled with a nitrobenzoxadiazole frame, which emits green-yellow fluorescence at 552 nm upon excitation at 413 nm, forming NBD1242 (Figure 3). A total of 1.5 × 10^4^ cells/well of CT26 tumor cells were seeded on collagen I-coated 12 mm cover slips within the 24-well plates. The cells were incubated at 37 °C for 24 h before the medium including NBD1242 (3 mM) was added. After incubation for 23 h, the supernatant was aspirated and BODIPY^®^ TR methyl Ester (Invitrogen, C34556, GFP Counterstain) 5 μM was added at 37 °C for 30 min. After incubation, the solutions were washed by culture medium and 1 μM Hoechst 33,342 (DOJINDO LABORATORIES, H342) stain was added at room temperature for 10 min. The sample solutions were washed by culture medium twice before a PBS 1× wash and fix with 4% paraformaldehyde at room temperature for 20 min. The cells were mounted on slides, incubated 3 times with PBS and ProLong Diamond (Invitrogen, P36961) for 5 min each, then stocked at 4 °C for 4 days. Samples were observed under a KEYENCE BZ-X710 (Keyence, Co. Ltd., Tokyo, Japan) fluorescence microscope.

### 2.3. Intracellular Uptake of MMT1242 In Vitro

To evaluate intracellular uptake of natural boron after injection of MMT1242, several types of cells, including B16-F10 mouse melanoma cells, C6 rat glioma cells, and CT26 mouse colon tumor cells were used. 1 × 10^6^ cells per well were seeded in 24-well plates. After 24 h of pre-incubation, we added 3 mM MMT1242 for each group (*n* = 3). Each cell line was incubated in either RPMI 1640 (Gibco A10491), Dulbecco’s Modified Eagle’s Medium (Gibco 11995-065), or F-12K Medium (Kaighn’s Modification of Ham’s F-12 Medium, ATCC 30-2004), as appropriate, together with fetal bovine serum and penicillin-streptomycin. Concentrations of natural boron were calculated by the ICP-MS (ICP-MS7700x, Agilent Technologies, California, CA, USA) method. The temperature dependence of intracellular boron uptake in CT26 tumor cells were also determined. A total of 1 mM each of MMT1242 and BPA were added to medium containing CT26 tumor cells at 37 °C and 4 °C for 1 h up to 3 h. Then, the intracellular concentration of natural boron was determined by the ICP-MS method. 

Then, to determine if uptake of MMT1242 was GLUT1 receptor dependent, we added cytochalasin B or high glucose. 10^6^ cells per well were seeded in 24-well plates. After 24 h of pre-incubation, we added 3mM MMT1242 for each group (*n* = 3). Each group contained either control medium (low glucose), low glucose with cytochalasin B (50 μM), high glucose, or high glucose with cytochalasin B (50 μM). Low glucose medium contained 1 g glucose in 1 L medium (for making low glucose medium, d-glucose was added to glucose-free 11879-RPMI 1640, GIBCO) and high glucose medium contained 4.5 g glucose in 1 L (RPMI 1640, GIBCO A10491). After 23 h incubation, cells were counted. Concentrations of natural boron were calculated by the ICP-MS method. Data were analyzed by one-way ANOVA and then differences (*p* < 0.05) among means of control (low glucose), low glucose with cytochalasin B, high glucose, and high glucose with cytochalasin B groups were analyzed using Tukey-Kramer multiple comparison tests.

### 2.4. Toxicity Study Using Healthy Mice

To check whether MMT1242 is toxic or not in vivo, a toxicity study using normal, healthy mice was done. Fourteen-week-old, female balb/c mice were given MMT1242 at 210 mg/kg in either a single intravenous injection (iv) group (*n* = 3) or 250 mg/kg group compared to a control (PBS) iv group (*n* = 3). Mice were serially weighed for 7 days. Renal specimens were taken after sacrifice for boron concentration as measured by ICP-MS at days 1, 3, and 7. Tissue specimens (from kidneys, hearts, lungs livers, and spleens) were evaluated on day 7.

### 2.5. In Vivo Fluorescence Imaging of ^10^B Distribution in Tumor Tissues

To observe ^10^B distribution in the tumor tissues of an animal model, 3 × 10^6^ mouse colon carcinoma (CT26) cells were injected in the right thigh of 4-week-old female BALB/c mice (In-Vivo Science, Inc., Tokyo, Japan). Two weeks after injection, tumor-bearing (avg. 1880 mm^3^) mice were used in this study. MMT1242 (250 mg/kg with 68.3 mg of [^10^B]/kg, *n* = 1) was administered by tail vein injection 24 h before sampling tissues. BSH (100 mg/kg with 57.1 mg of [^10^B]/kg, *n* = 1) and BPA (500 mg/kg with 24.1 mg of [^10^B]/kg, *n* = 1) were administered by tail vein injection 2 h before tissue sampling. PBS (*n* = 1) was administered by tail vein injection 2 h before sampling tissues as a control. Samples were embedded in Tissue-Tek^®^ O.C.T. Compound (Sakura Finetek Japan Co., Ltd., Tokyo, Japan) and frozen at −70 °C by acetone cooled with dry ice and sliced to a thickness of 10 μm (New Histo. Science Laboratory Co., Tokyo, Japan). These serial sections were used for ^10^B distribution evaluation by LA-ICP-MS and HE staining, respectively. We used an NWR213 laser ablation (LA) system equipped with a two-volume ablation chamber (ESI, Portland, OR, USA) coupled with an inductively coupled plasma sector field mass spectrometer (ICP-SF MS; Element XR, Thermo Fisher Scientific, Bremen, Germany). The applied mass spectrometer scanning conditions contained ^10^B which were used for data acquisition with an integration time of 2 ms. The raw data of an LA-ICP-MS image consisted of a sequence of line scans. To visualize the raw data, the line scans for each element were transformed into a matrix (text image) using a customized Excel-macro program. This matrix was visualized using iQuant2, which was developed by Suzuki et al. [47].

### 2.6. Biodistribution of Boron Compounds In Vivo

To clarify the biodistribution of natural boron after injection of MMT1242 in our animal model, CT26 mouse colon tumor cells were subcutaneously injected into the lumbar tissues of the mice. Each boron compound was injected into these model animals. BPA, BSH, and MMT1242 dosages and their natural boron content at each dose were 500 mg/kg (26 mg[B]/kg, *n* = 3) for BPA (with 2.6 equivalents of fructose), 100 mg/kg (59 mg[B]/kg, *n* = 3) for BSH, and 210 mg/kg (59 mg[B]/kg, *n* = 3) for MMT1242.

For BPA and BSH, the tumor tissues, livers, kidneys, and blood were collected while for MMT1242 the tumor tissue, liver, kidney, blood, thigh muscle, brain and spleen were collected after 1 h, 3 h, 6 h, and 24 h. All samples, except blood samples, weighed between 20 and 50 mg and were frozen and crushed. The 0.1 mL blood samples were dissolved in aqueous nitric acid solution and boiled for 1 h at 60 °C before boiling for 2 h at 90 °C. After filtration of insoluble matter, all samples were diluted 20 times with deionized water before determination of natural boron concentration by ICP-MS. Tumor-to-blood ratios (T/B ratio) were calculated and compared among BPA, BSH, and the MMT1242 groups.

### 2.7. Irradiation Study Using a Mouse Tumor Model

To assess the suitability of MMT1242 for BNCT, we injected 3 × 10^6^ mouse colon carcinoma (CT26) cells into the right thighs of 4-week-old female BALB/c mice (In-Vivo Science, Inc., Tokyo, Japan). Two weeks after injection, tumor-bearing (avg. 360 mm^3^) mice were used in this study. Two doses of ^10^B-MMT1242 (88.8 mg/kg with 24.1 mg[^10^B]/kg, *n* = 5 and 250 mg/kg with 68.3 mg[^10^B]/kg, *n* = 6) were administered by tail vein injection 24 h before irradiation. Fructose-^10^B-BPA (500 mg/kg with 24.1 mg[^10^B]/kg, *n* = 5) was administered by tail vein injection 2 h before irradiation as a positive control. Irradiation only (*n* = 4) and untreated mice (*n* = 6) were used as controls. The irradiation was performed with thermal neutrons with a flux of 1.8–4.0 × 10^12^ neutrons/cm^2^ over 1 h at the Kyoto University Research Reactor (KUR). Tumor sizes and body weights were measured in the period starting from before the treatment until 26 days after irradiation. Tumor volumes (mm^3^) were calculated using the following Equation (1):(1)$$Tumor\;Volume=\frac{Long\;Diameter\times Short\;Diameter^{2}}{2}$$

Results are expressed as means of relative increase ratio of tumor volume as follows (Equation (2)): (2)$$Relative\;Increase\;Ratio\;of\;Tumor\;Volume=\frac{Tumor\;Volume\;at\;Day\;26}{Tumor\;Volume\;at\;Day\;0}$$

The statistical differences between no treatment and irradiation only groups were determined by two-sided Student’s *t*-test. Difference with *p* < 0.05 was considered significant. Data were analyzed by one-way ANOVA, and then differences (*p* < 0.05) among means of irradiation only, BPA, ^10^B-MMT1242 (89 mg/kg), and ^10^B-MMT1242 (250 mg/kg) groups were analyzed using Tukey-Kramer multiple comparison tests.

## 3. Results

MMT1242 demonstrated both a reproducibly high uptake in all tested cell lines in vitro and a sufficient tumor to normal tissue accumulation ratio for BNCT in biodistribution studies. Our murine subcutaneous tumor model irradiation study showed that MMT1242 had a significant tumor-inhibiting effect with the injection 24 h prior to neutron irradiation.

### 3.1. Intracellular Distribution of MMT1242 In Vitro

Fluorescence microscopy revealed the intracellular distribution of NBD-labeled MMT1242 (NBD1242) in CT26 tumor cells as diffuse and broad (Figure 4).

### 3.2. Intracellular Uptake of Boron MMT1242 In Vitro

#### 3.2.1. Intracellular Uptake of Boron in CT26 Mouse Colon Tumor Cells

Compared to BPA, twice as much MMT1242 was uptaken by CT26 tumor cells than BPA (Figure 5a). MMT1242 also tended to be uptaken by CT26 tumor cells in a dose-dependent manner (Figure 5a).

#### 3.2.2. Intracellular Uptake of Boron in the B16-F10 Mouse Melanoma Cells/C6 Rat Brain Tumor Cells

The 3 mM MMT1242 dosage in B16-F10 melanoma cells resulted in a 2.7× higher concentration of boron compared to BPA while the same doses in C6 glioma cells resulted in 8× more accumulated boron than BPA. MMT1242 tended to be uptaken by several different types of tumor cells more than BPA, which was poorly accumulated in C6 glioma cells compared to B16-F10 and CT26 tumor cells (Figure 5b). Morever, MMT1242 was uptaken by all three cell lines adequately (Figure 5b).

#### 3.2.3. Retention of Intracellular Boron in CT26 Tumor Cells

The intracellular BPA concentration decreased immediately in a time-dependent manner when cells were washed (Figure 5c). However, the high intracellular MMT1242 concentration was maintained up to 60 min in tumor cells.

#### 3.2.4. Temperature Dependence of Intracellular Uptake of Boron in CT26 Tumor Cells

Intracellular boron concentrations increased in a time-dependent manner at 37 °C but not at 4 °C for both compounds (Figure 5d). After adding MMT1242, the intracellular boron concentrations in the CT26 tumor cells were 2.7× higher than BPA after 1 h incubation and 8.8× higher than BPA after 3 h incubation. These results show that MMT1242 is uptaken by tumor cells with active transport (endocytosis) at 37 °C, but this process is inhibited at 4 °C.

#### 3.2.5. Intracellular Boron Uptake in CT26 Tumor Cells by GLUT1 Inhibition

Neither high glucose nor cytochalasin B inhibited the uptake of MMT1242 in CT26 tumor cells (Figure 5e). Paradoxically, low glucose with cytochalasin B resulted in a significantly higher boron concentration, suggesting a GLUT1 receptor-independent mechanism of MMT1242 accumulation in tumor cells.

### 3.3. Toxicity Study Using Healthy Mice

The weights of all mice did not change for 7 days after intravenous injection of MMT1242 (Figure 6a). MMT1242 showed good clearance from normal kidneys in mice (Figure 6b). No toxic damage was observed in tissue specimens of kidney, heart, lung, liver, or spleen from mice (Figure 6c).

#### 3.3.1. In Vivo Fluorescence Imaging of ^10^B Distribution in Tumor Tissues

Before the animal irradiation study, boron distribution from MMT1242 and other ^10^B boron compounds in tumor tissue was analyzed by fluorescence imaging (Figure 7). In the case of ^10^B-BPA, boron was distributed heterogeneously within the tumor tissues due to tumor cellularity (Figure 7c), whereas ^10^B-BSH saw its boron distributed mainly in the interstitial spaces between the tumor tissues (Figure 7b). For ^10^B-MMT1242, the ^10^B was distributed highly within the tumor tissues together with a mild boron distribution in the interstitial spaces (Figure 7d). Distribution of ^10^B with ^10^B-MMT1242 had a broader range than that of ^10^B-BPA, which is more suitable for BNCT than the usual compounds BSH and BPA.

#### 3.3.2. Biodistribution of Boron Compounds In Vivo

The biodistribution of boron compounds is shown in Figure 8. The T/B ratio was 1.3 at 1 h, 9.9 at 3 h, 1.7 at 6 h, and 5.8 at 24 h for BSH. The T/B ratio was 2.8 at 1 h, 3.7 at 3 h, 3.6 at 6 h, and 7.8 at 24 h for BPA. The T/B ratio was 0.5 at 1 h, 0.6 at 3 h, 0.8 at 6 h, and 3.5 at 24 h for MMT1242. The boron concentration (μg B/g tissue) of BSH and BPA decreased over time but MMT1242 maintained high values even at 24 h after injection (BSH for 1.2 ppm, BPA for 1.2 ppm, and MMT1242 for 34.5 ppm at 24 h, respectively). These results demonstrate a superior retention of MMT1242 in the tumor cells compared to BPA.

#### 3.3.3. Irradiation Study Using Mouse Subcutaneous Tumor Model

We confirmed that MMT1242 has adequate intratumor tissue retention together with good clearance from normal tissue during in vivo ^10^B biodistribution studies. Thus, we did an irradiation study using a mouse subcutaneous CT26 tumor model. The results of the compound toxicity and animal tumor growth after neutron irradiation is shown in Figure 6. No effect of treatment on body weight for all individual mice was found (Figure 9a). The irradiation-only groups saw significantly suppressed tumor volumes compared with no-treatment groups (*p* = 0.0321). The ^10^B-MMT1242 (250 mg/kg) (*p* = 0.0123), ^10^B-MMT1242 (89 mg/kg) (*p* = 0.0002), and ^10^B-BPA (*p* < 0.0001) groups saw significant tumor volume suppression compared with irradiation-only groups (Figure 9b). The BPA groups also showed significant tumor suppression compared with ^10^B-MMT1242 (89 mg/kg) groups (*p* = 0.0171) (b). In particular, the ^10^B-MMT1242 (250 mg/kg) group showed a tumor growth inhibitory effect comparable to the ^10^B-BPA group up until the 18th day. No embolisms in the lung, spleen, or liver were observed over the study course.

## 4. Discussion

Based on these results, we report that ^10^B-MMT1242 is a suitable boron compound candidate for further investigation that showed high tumor accumulation even 24 h after injection. The main feature of MMT1242 is a mannose-based chemical structure, which is suitable for promoting active transport across cellular membranes. Moreover, the molecular weight of MMT1242 is low enough to make structural modification easy for developing a drug delivery system. The uptake and retention properties of the mannose-based structure thus make MMT1242 an attractive boron compound for further BNCT studies.

The ideal boron compound for BNCT selectively accumulates only in tumor cells at a high tumor to normal cells ratio by using active instead of passive transport systems. Mannose is phosphorylated by hexokinase (HK) intracellularly to produce mannose-6-phosphate (Man-6-P). It is either catabolized by phosphomannose isomerase (MPI) or directed into *N*-glycosylation via phosphomannomutase (PMM2). As malignant glioma cells and other cancer cells favor anaerobic conditions that promote high lactic acid production and hypoglycemia (Warburg effect) [Tran 2016], we expected upregulation of active glucose transporter (GLUT) and lactate dehydrogenase expression in these cells might favor MMT1242 accumulation. Moreover, as our novel boron compound contains α-d-mannopyranoside, we hypothesized an interaction with GLUT. However, our results indicated that MMT1242 uptake may be GLUT1 independent as inhibition of GLUT1 receptor by cytochalasin B or downregulation of GLUT1 receptor by high glucose did not affect the uptake of MMT1242.

Other possible mechanisms, such as endocytosis by mannose receptors on the plasma membrane of CT 26 mice colon carcinoma cells, may explain this result. Mannose receptors (MRs) are ubiquitously expressed on various types of cells [22] and murine mannose receptor CD206 (Cluster of Differentiation 206) expression in mouse colon was previously observed [48]. Mannose receptors are also expressed in CT26 colon tumor cells [31] and strongly correlate with tumorigenesis in the interstitial space of many types of cancers [49,50,51,52], in addition to the Warburg effect [49,53,54,55]. Inspite of significant long-tailed mannopyranoside modifications, including *N*-(4-nitrobenzo-2-oxa-1,3-diazole) (NBD), these compounds were also selectively uptaken by mannose receptors [30]. Thus, the synthesis of mannosylated constructs can contribute to the active transport of boronated compounds into many types of tumor cells for effective targeted delivery of BNCT-compatible bioactive agents. Further uptake mechanisms should be elucidated by future studies.

BNCT is an ideal candidate for tumoricidal therapy as it can treat not only malignant glioma, melanoma, head and neck cancers, gingival cancer, and lung cancer but also recurrent malignant brain tumors and many types of cancers. As most of these tumors together with the Warburg effect, our novel compound, MMT1242, might be suitable for future BNCT studies in these types of tumors.

Both the MMT1242 in vitro and in vivo studies showed high tumor to blood ratios even at 24 h after administration. Compared to BPA, the distribution of MMT1242 in the tumor tissues was broader, including within the interstitial space of tumor tissues. With regard to biological persistence of BSH and BPA, MMT1242 might have some advantage to maintaining high T/N or high T/N ratios in tumor tissues beyond 24 h. However, concerns about slightly high blood MMT1242 concentrations in the animal model may be explained by heterogenous distribution due to aggregation with macroproteins. This could be causative of the poorer tumor growth inhibiting effect seen in the MMT1242 group compared to the BPA group in the irradiation study. Aggregated MMT1242 might readily attach to the glycoproteins on the cell membranes as previous reports indicate that mannose-related compounds broadly attach to diverse glycoproteins on the plasma membranes of human colon carcinoma cell lines [29,56]. Another reason could be diffusion of MMT1242 into the interstitial space within the tumor tissue. Both possibilities would lead to aggregation of MMT1242 on the surface of the cell membrane and not in the tumor tissue cytosol. In either case, slight overestimation of intracellular boron distribution might occur. Another reason why MMT1242 showed increased retention in tumor cells might be its highly negative charge. Some compounds showed a similar effect in previous studies [57,58]. However, the precise mechanisms of MMT1242 uptake into the tumor cells are still unknown and, as real-time monitoring of this process is impossible, conjugation with nanoparticles or PEGylation of liposomes is mandatory. Fortunately, with the lower molecular weight of its chemical structure, such modifications (and many others) are possible and easier to accomplish. On the other hand, increasing the tumor to normal retention ratio with BPA means that, in BNCT clinical trials, BPA had to be continuously administered to malignant glioma patients just prior to neutron irradiation. Early clearance of BPA from the tumor tissues is definitely a problem associated with standard BPA [59,60].

Thus, although the precise mechanism is unknown, ^10^B-MMT1242 is a promising novel candidate for all-around performance in BNCT, as its effects do not seem dependent on the differences between tumor cells. Therefore, ^10^B-MMT1242 might be a potential ^10^B-boron compound against the many malignant tumor cells susceptible to BNCT.

Regarding biocompatibility, MMT1242 is a mannosylated-saccharide based compound, and a family of glucose transporters recognizes and mediates passage of glucose and substances exhibiting similar structures, including 2-deoxyglucose, galactose, mannose, and glucose analogs, through the blood brain barrier [25]. Mannose is absorbed through the intestine, metabolized, and cleanly washed out within 4 h without affecting glucose concentrations [22]. This is a key point, as other artificial boron-containing compounds with much higher tumor to normal tissue retention ratios carried risks of pulmonary and splenic embolism, liver dysfunction, and severe renal dysfunction in preliminary animal biodistribution and toxicity studies [61,62,63]. In contrast, our compound caused no decrease in mice weight while liver and kidney specimens were free from embolisms and showed no particular toxic effect. Although more toxicity studies are needed, MMT1242 might be an ideal, natural boron-containing compound for effective BNCT.

There are limitations to this study. First, the cutoff for a low-molecular-weight-boron compound is usually defined as below 10^4^ [64]. Therefore, the MMT1242 molecular weight is low but, as it is higher than that of BPA, we have to consider the possibility of different biodistribution and tumor uptake dynamics among MMT1242, BPA, and BSH due to their different molecular weights in future studies. Second, although the amounts of natural B and ^10^B in mg/kg equivalents administered in the form of BPA or MMT1242 in the biodistribution study were not the same, we chose only those doses of MMT1242 that were non-toxic and showed both good retention in tumor cells and adequate tumor-to-normal tissue accumulation ratio. According to the results of the biodistribution study, we did irradiation experiments using a low and a high dose of MMT1242 compared to BPA. Although the amounts of administered ^10^B were not equivalent in mg/kg to the high dose of MMT1242 and BPA in the irradiation study, the high dose of MMT1242 showed low toxicity and significant tumor inhibiting effect upon injection 24 h before irradiation compared to the low doses of MMT1242 and BPA. Third, for human application of MMT1242 in BNCT, more precise toxicity studies based on animal experiments are needed. Although we showed MMT1242 to have a broad and retentive distribution in diverse cancer cell lines and tissues, more precise studies on uptake and retention mechanisms in the tumor cells need to be evaluated before stepping up to human clinical trials.

## 5. Conclusions

Our novel boron-containing α-d-mannopyranoside MMT1242 demonstrated a high tumor to normal tissue accumulation ratio in the tumor models and also clearly inhibited tumor growth even 24 h after administration in the animal irradiation studies. MMT1242 is thus a promising candidate for future accelerator-based BNCT clinical trials.

## 6. Patents

A patent application was filed with the Japan Patent Office (JPO).

## Figures and Tables

**Figure 1 cells-09-01277-f001:**
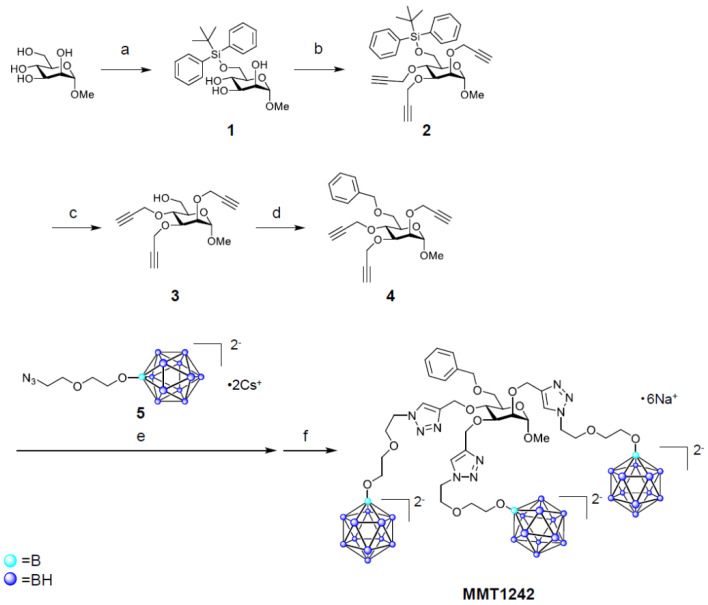
Synthesis of MMT1242. Reaction conditions and yields: (**a**) *tert*-butylchlorodiphenylsilane, imidazole, *N*,*N*-dimethylformamide, 5 °C to rt (quant.); (**b**) sodium hydride, *N,N*-dimethylformamide, 5 °C to rt, then propargyl bromide, cat. tetrabutylammonium iodide, 5 °C to rt (68%); (**c**) tetrabutylammonium fluoride, tetrahydrofuran, 5 °C to rt (91%); (**d**) sodium hydride, *N,N*-dimethylformamide, 5 °C to rt, then benzyl bromide, rt (53%); (**e**) **5**, copper (II) sulfate pentahydrate, sodium l-ascorbate, *tert*-butanol/H_2_O (1/2, *v*/*v*), 70 °C; (**f**) Amberlite^®^ IR120B Na, H_2_O, (25%, 2 steps).

**Figure 2 cells-09-01277-f002:**
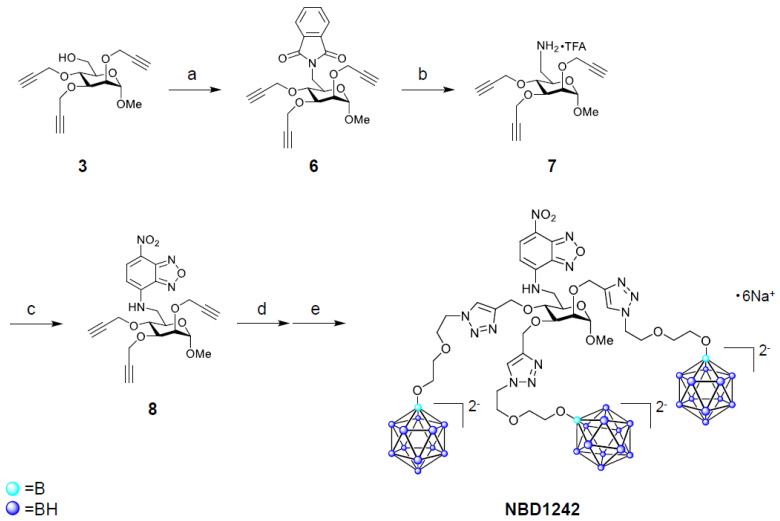
Synthesis of NBD1242. Reaction conditions and yields: (**a**) phthalimide, triphenylphosphine, diethyl azodicarboxylate (40% in toluene), tetrahydrofuran, 5 °C to rt; (**b**) hydrazine monohydrate, ethanol, rt, then preparative RP-HPLC (containing 0.05% trifluoroacetic acid), (50%); (**c**) 4-fluoro-7-nitro-2,1,3-benzoxadiazole, triethylamine, *N*,*N*-dimethylformamide, 50 °C (38%); (**d**) **5**, copper (II) sulfate pentahydrate, sodium l-ascorbate, *tert*-butanol/H_2_O (1/2, *v*/*v*), 70 °C; (**e**) H_2_O, Amberlite^®^ IR120B Na (19%, 2 steps).

**Figure 3 cells-09-01277-f003:**
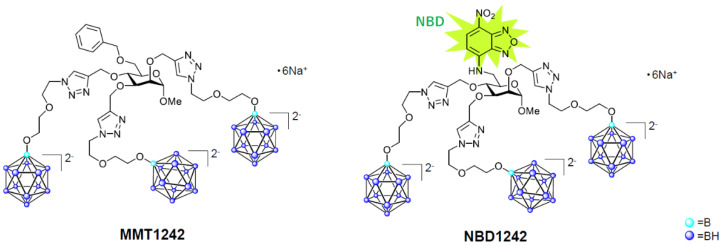
The chemical structures of MMT1242 and NBD1242.

**Figure 4 cells-09-01277-f004:**
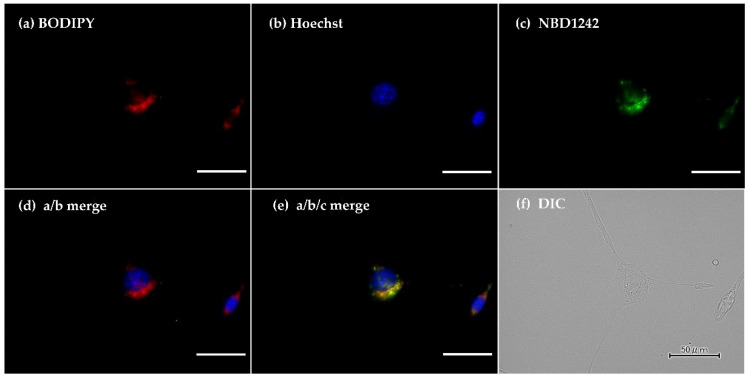
The intracellular distribution of NBD1242 in CT26 colon tumor cells. **a** (BODIPY), **b** (DAPI), **c** (NBD1242), **d** (merge 4a with 4b), **e** (merge 4a with 4b and 4c), and **f** (DIC). Fluorescence microscopy showing intracellular distribution of NBD1242. The scale bar is 50 μm.

**Figure 5 cells-09-01277-f005:**
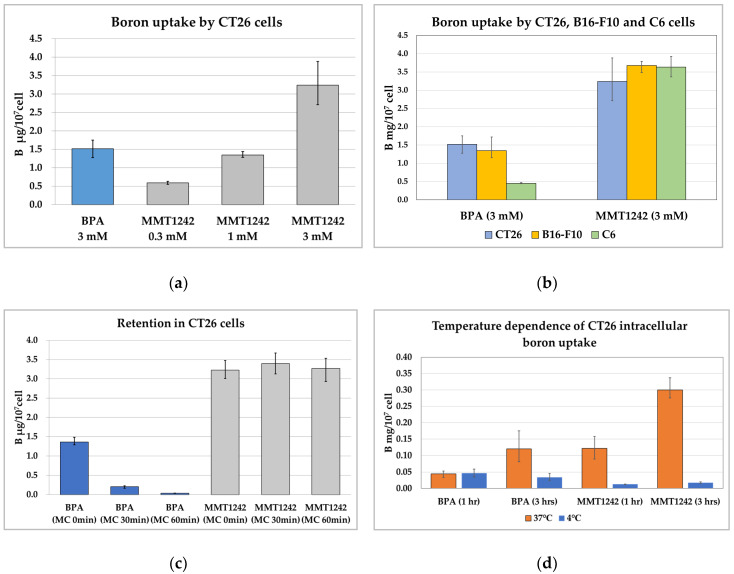
The intracellular uptake of MMT1242 among the different kinds of tumor cells lines. The intracellular uptake of MMT1242 resulting in high boron concentrations in the CT26 tumor cells (**a**), and in the different kinds of tumor cell lines (**b**). MMT1242 showing intracellular retention in CT26 tumor cells (**c**) The temperature dependence of intracellular boron uptake in CT26 tumor cells (**d**). Intracellular boron uptake in CT26 tumor cells by GLUT1 inhibition (**e**).

**Figure 6 cells-09-01277-f006:**
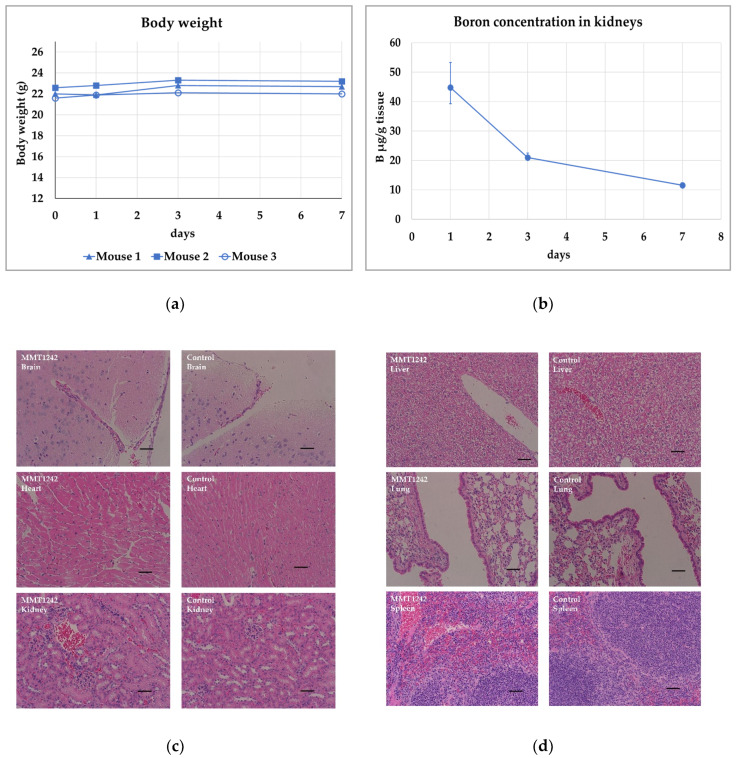
Toxicity study using healthy mice. All mouse weights were unchanged for 7 days after intravenous injection of MMT1242 (**a**). The boron concentration in mouse kidneys showed good clearance within 7 days after injection of MMT1242 (**b**). HE-stained tissue specimens of kidney, heart, lung, liver, and spleen from mice showing no toxic damage (**c**,**d**). The scale bar is 50 μm.

**Figure 7 cells-09-01277-f007:**
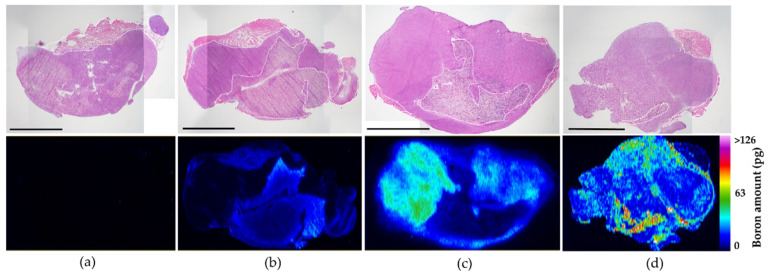
In vivo fluorescence imaging of ^10^B distribution of ^10^B-MMT1242 in tumor tissues. An in vivo ^10^B distribution study showing slightly heterogeneous but broad range of ^10^B distribution in tumor tissues after ^10^B-MMT1242 administration (**d**) compared to control (**a**), ^10^B-BSH (**b**), and ^10^B-BPA (**c**). Each black scale bar stands for 5mm. Each white dotted line in the figures represents interstitial spaces in the tumor tissues.

**Figure 8 cells-09-01277-f008:**
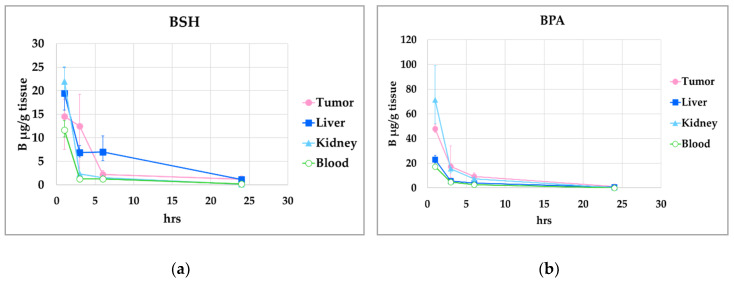
The biodistribution of MMT1242 in mice tumor models. Higher tumor-to-blood ratio as seen 24 h after injection of MMT1242 (**c**) compared with BPA (**b**) or BSH (**a**).

**Figure 9 cells-09-01277-f009:**
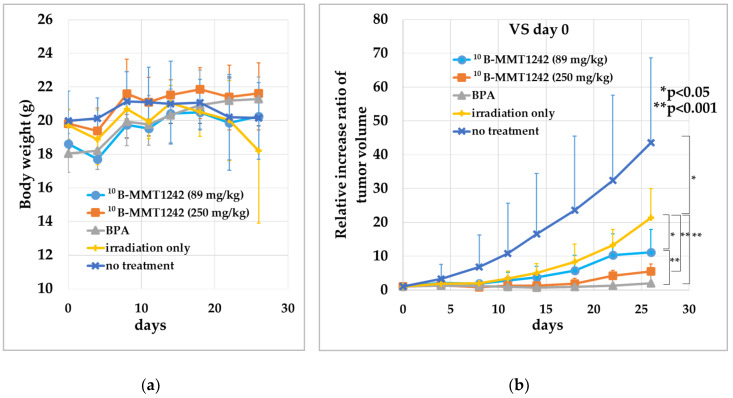
BNCT of tumor-bearing mice with ^10^B-enriched compounds. (**a**) Mice body weight after thermal neutron irradiation. (**b**) Tumor growth ratio after 1-h epithermal neutron irradiation (1.8–4.0 × 10^12^ neutrons/cm^2^) after the injection of ^10^B-MMT1242 at 24 h and ^10^B-BPA at 2 h before irradiation with irradiation-only and untreated groups as controls. Comparisons by t-test for no-treatment and irradiation-only groups, and comparisons by Tukey-Kramer multiple comparison tests for irradiation-only, BPA, ^10^B-MMT1242 (89 mg/kg), and ^10^B-MMT1242 (250 mg/kg) groups were performed using mice 26 days after irradiation.

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
