# Peer review of "Evaluation of a Novel Boron-Containing α-d-Mannopyranoside for BNCT"

_cells, 2020, doi:10.3390/cells9051277_

Round 1

Reviewer 1 Report

Boron neutron capture therapy (BNCT) is a promising method for treatment of cancer based on the selective accumulation of the non-radioactive isotope 10B in tumor cells and their subsequent treatment with low-energy thermal neutrons. This method is developed now in several laboratories. Scientists try to find different way to solve a problem. Perspective methods are to prepare conjugates of boron clusters with 10 or more boron atoms in molecule with biomolecules that help to selective delivery boron species into tumor cell. That is why it is important a purpose of this work to develop a novel boron-containing α-D-mannopyranoside,  for BNCT. Authors presented important complex investigation that concludes chemical synthesis of new compounds,  intracellular distribution and retention of MMT1242 in culture cells, and analyze biodistribution, tumor-to-normal tissue ratio and toxicity in vivo.

A  neutron irradiation study with MMT1242 in a subcutaneous murine tumor model showed low  toxicity and significant tumor inhibiting effect upon injection 24 hours before irradiation. That permit authors to report that 10B-MMT1242 is a candidate for further investigation in clinical BNCT studies.

Based on these results it is possible to conclude that this paper may be published after several corrections.

Remarks:

It is not clear from the text, if boron compounds used in the work, had a natural content of boron, or author used 10B isotope. In last case what was the source of this compounds.

On page 2, lines 92-93 it is written that compound 5 was prepared according to the methods described in the patent (WO2017164334).  However this compound was first prepared and described in 2007 (A.V.Orlova, N.N.Kondakov, B.G.Kimel, L.O.Kononov, E.G.Kononova, I.B.Sivaev, V.I.Bregadze. Synthesis of novel derivatives of closo-dodecaborate anion with azido group at the terminal position of the spacer. Appl. Organomet. Chem, 2007, 21, №2, 98-100.). This paper should be mentioned, and the difference of two methods should be discussed.

In introduction, in order to give main principles of BNCT and present last results in this field it is necessary to include to Introduction some reference on last monograph and reviews, for example:

  1. Neutron Capture Therapy. Principles and Applications (Eds. W. Sauerwein, A. Wittig, R. Moss, Y. Nakagawa), Springer-Verlag, Berlin, 2012.
  2. I. B. Sivaev, V. I. Bregadze, Eur. J. Inorg. Chem. 2009, 11, 1433-1450.
  3. R. F. Barth, Z. Zhang, T. Liu, A realistic appraisal of boron neutron capture therapy as a cancer treatment modality, Cancer Commun., 2018, 38, 36                 Since the main subject of this manuscript is boronated carbohydrates current reviews and main  papers on this subject should be cited:

1.M. Kotora, Carboranyl-saccharide Derivatives: Syntheses and Biological Evaluation, In: Handbook of Boron Science with Applications in Organometallics, Catalysis, Materials and Medicine. Vol. 4. Boron in Medicine (Eds. N. S. Hosmane, R. Eagling), World Scientific: London, 2018, pp. 69-99.

  1. A.V. Orlova, L. O. Kononov, Synthesis of conjugates of polyhedral boron compounds with carbohydrates, Russ. Chem. Rev., 2009, 78, 629-648.

3.R. Satapathy, B. P. Dash, C. S. Mahanta, B. R. Swain, B. B. Jena, N. S. Hosmane, Glycoconjugates of polyhedral boron clusters, J. Organomet. Chem., 2015, 798, 13-23.

Author Response

Boron neutron capture therapy (BNCT) is a promising method for treatment of cancer based on the selective accumulation of the non-radioactive isotope 10B in tumor cells and their subsequent treatment with low-energy thermal neutrons. This method is developed now in several laboratories. Scientists try to find different way to solve a problem. Perspective methods are to prepare conjugates of boron clusters with 10 or more boron atoms in molecule with biomolecules that help to selective delivery boron species into tumor cell. That is why it is important a purpose of this work to develop a novel boron-containing α-D-mannopyranoside, for BNCT. Authors presented important complex investigation that concludes chemical synthesis of new compounds, intracellular distribution and retention of MMT1242 in culture cells, and analyze biodistribution, tumor-to-normal tissue ratio and toxicity in vivo.

A neutron irradiation study with MMT1242 in a subcutaneous murine tumor model showed low toxicity and significant tumor inhibiting effect upon injection 24 hours before irradiation. That permit authors to report that 10B-MMT1242 is a candidate for further investigation in clinical BNCT studies.

Response: We thank the reviewer for positive comments and helpful suggestions. We have taken all these comments and suggestions into account and they have improved our manuscript considerably.

Based on these results it is possible to conclude that this paper may be published after several corrections.

Response:  We thank the reviewer for positive comments and helpful suggestions. We have taken all these comments and suggestions into account and they have improved our manuscript considerably.

Remarks:

Reviewer: It is not clear from the text, if boron compounds used in the work, had a natural content of boron, or author used 10B isotope. In last case what was the source of this compounds.

Response:  We have added the following sentence to Materials and Methods:

“MMT1242 containing 10B isotope was used in fluorescence imaging of 10B distribution in vivo and in irradiation experiments while, in all other experiments, MMT1242 containing natural boron was used.”

Reviewer: On page 2, lines 92-93 it is written that compound 5 was prepared according to the methods described in the patent (WO2017164334).  However this compound was first prepared and described in 2007 (A.V.Orlova, N.N.Kondakov, B.G.Kimel, L.O.Kononov, E.G.Kononova, I.B.Sivaev, V.I.Bregadze. Synthesis of novel derivatives of closo-dodecaborate anion with azido group at the terminal position of the spacer. Appl. Organomet. Chem, 2007, 21, №2, 98-100.). This paper should be mentioned, and the difference of two methods should be discussed.

Response:  We have added the following information to Materials and Methods:

“The closo-dodecaborate unit (5) that was previously synthesized by Orlova et al. (2007) and Semioshkin et al. (2007) was here prepared according to our methods described in the patent (WO2017164334) and, according to our modifications, contains a cesium salt to provide sufficient water solubility in contrast to the previously reported compound that contained a tetrabutylammonium salt [43,44].”

  1. Orlova, A.V.; Kondakov, N.N.; Kimel’, B.G.; Kononov, L.O.; Kononova, E.G.; Sivaev, I.B.; Bregadze, V.I. Synthesis of novel derivatives of closo-dodecaborate anion with azido group at the terminal position of the spacer. Appl Organomet Chem 2007, 21, №2, 98-100. DOI: 10.1002/aoc.1151.
  2. Semioshkin, A.A.; Osipov, S.N.; Grebenyuk, J.N.; Nizhnik, E.A.; Godovikov, I.A.; Shchetnikov, G.T.; Bregadze, V.I. An Effective Approach to 1,2,3-Triazole-Containing 12-Vertex closo-Dodecaborates. Collect Czech Chem Commun 2007, 72, 1717-1724. DOI: 10.1135/cccc20071717.

Reviewer: In introduction, in order to give main principles of BNCT and present last results in this field it is necessary to include to Introduction some reference on last monograph and reviews, for example:

  1. Neutron Capture Therapy. Principles and Applications (Eds. W. Sauerwein, A. Wittig, R. Moss, Y. Nakagawa), Springer-Verlag, Berlin, 2012.
  2. I. B. Sivaev, V. I. Bregadze, Eur. J. Inorg. Chem. 2009, 11, 1433-1450.
  3. R. F. Barth, Z. Zhang, T. Liu, A realistic appraisal of boron neutron capture therapy as a cancer treatment modality, Cancer Commun., 2018, 38, 36                 

Response: We have modified the following part of the Introduction and added the mentioned authors in the References here and further in the text.

“Boron neutron capture therapy (BNCT) is a binary anticancer technology that was first proposed by Locher in 1936 and has since been further developed and improved by a number of research groups all over the world. It commands attention as a state-of-the-art, minimally invasive cancer treatment that functions as a single cell level-selective tumoricidal therapy [1,2,3,4,5,6,7]. It employs a stable boron-10 (10B) isotope-containing compound injected into the patient and neutron irradiation produced at a nuclear reactor (or accelerator) and, once the appropriate form of 10B-boron-containing compound has selectively accumulated in malignant tumor cells by a drug delivery system (DDS), the subsequent neutron irradiation of the tumor area induces alpha particles and Li nuclei emission within a 10 micrometer span, limiting the BNCT effect to tumor cells with accumulated boron and sparing the surrounding normal cells without boron accumulation [2,3,4,8,9]. This tumor-selective, high-dose irradiation may improve local control and survival after the initial treatment. Phase I/II clinical trials of BNCT were conducted for patients with glioblastoma and the early results suggest that BNCT is effective in selected cases [10,11,12].

Until recently, clinical studies using boronophenylalanine (BPA) and sodium borocaptate (BSH) as 10B delivery agents for the neutron capture reaction have been conducted [3,8,10,13]. However, continuous administration at high concentrations is needed to maintain sufficient 10B concentration within tumors [14]. For effective BNCT, ≥ 20 μg of 10B per gram of tumor must be selectively delivered to the tumor cells (~109 atoms per cell), reside intracellularly for a sufficient amount of time, and simultaneously clear from surrounding normal tissues to ideally attain a tumor-to-normal tissue ratio of 3-4 to 1 [2,3,4,5,15,16].”

  1. Locher G. Biological effects and therapeutic possibilities of neutrons. Am J Roentgenol Radium Ther 1936, 36, 1-13.
  2. Soloway, A.H.; Tjarks, W.; Barnum, B.A.; Rong, F.G.; Barth, R.F.; Codogni, I.M.; Wilson, J.G. The chemistry of neutron capture therapy. Chem Rev 1998, 98, 1515–1562. DOI: 10.1021/cr941195u.
  3. Sauerwein, W.A.G.; Wittig, A.; Moss, R.; Nakagawa, Y. Neutron Capture Therapy. Principles and Applications. Springer-Verlag: Berlin, Heidelberg, Germany. 2012.
  4. Sivaev, I.B.; Bregadze, V.I. Polyhedral boranes for medical applications: Current status and perspectives. Eur J Inorg Chem 2009, 11, 1433-1450. DOI: 10.1002/ejic.200900003.
  5. Barth, R.F.; Zhang, Z.; Liu, T. A realistic appraisal of boron neutron capture therapy as a cancer treatment modality. Cancer Commun (Lond) 2018, 38, 36. DOI: 10.1186/s40880-018-0280-5.
  6. Kreiner, A.J.; Bergueiro, J.; Cartelli, D.; Baldo, M.; Castell, W.; Asoia, J.G.; Padulo, J.; Suárez Sandín, J.C.; Igarzabal, M.; Erhardt, J.; Mercuri, D.; Valda, A.A.; Minsky, D.M.; Debray, M.E.; Somacal, H.R.; Capoulat, M.E.; Herrera, M.S.; Del Grosso, M.F.; Gagetti, L.; Anzorena, M.S.; Canepa, N.; Real, N.; Gun, M.; Tacca, H. Present status of Accelerator-Based BNCT. Rep Pract Oncol Radiother 2016, 21, 95-101. DOI: 10.1016/j.rpor.2014.11.004.
  7. Taskaev, S. Development of an accelerator-based epithermal neutron source for boron neutron capture therapy. Phys Part Nucl 2019, 50, 569–575. DOI: 10.1134/S1063779619050228.
  8. Barth, R.F.; Mi, P.; Yang, W. Boron delivery agents for neutron capture therapy of cancer. Cancer Commun(Lond) 2018, 38, 35. DOI: 10.1186/s40880-018-0299-7.
  9. Sato, E.; Zaboronok, A.; Yamamoto, T.; Nakai, K.; Taskaev, S.; Volkova, O.; Mechetina, L.; Taranin, A.; Kanygin, V.; Isobe, T.; Mathis, B. J.; & Matsumura, A. Radiobiological response of U251MG, CHO-K1 and V79 cell lines to accelerator-based boron neutron capture therapy. J Radiat Res 2018, 59, 101-107. DOI: 10.1093/jrr/rrx071.
  10. Yamamoto, T.; Nakai, K.; Tsurubuchi, T.; Matsuda, M.; Shirakawa, M.; Zaboronok, A.; Endo, K.; Matsumura, A. Boron neutron capture therapy for newly diagnosed glioblastoma: a pilot study in Tsukuba. Appl Radiat Isot 2009, 67(7-8 Suppl), S25-26. DOI: 10.1016/j.apradiso.2009.03.011.
  11. Kawabata, S.; Miyatake, S.; Hiramatsu, R.; Hirota, Y.; Miyata, S.; Takekita, Y.; Kuroiwa, T.; Kirihata, M.; Sakurai, Y.; Maruhashi, A.; Ono, K. Phase II clinical study of boron neutron capture therapy combined with X-ray radiotherapy/temozolomide in patients with newly diagnosed glioblastoma multiforme--study design and current status report. Appl Radiat Isot 2011, 69, 1796-1799. DOI: 10.1016/j.apradiso.2011.03.014.
  12. Miyatake, S.; Kawabata, S.; Hiramatsu, R.; Kuroiwa, T.; Suzuki, M.; Kondo, N.; Ono, K. Boron Neutron Capture Therapy for Malignant Brain Tumors. Neurol Med Chir (Tokyo) 2016, 56, 361-371. DOI: 10.2176/nmc.ra.2015-0297.
  13. Wang, L.W.; Chen, Y.W.; Ho, C.Y.; Hsueh Liu, Y.W.; Chou, F.I.; Liu, Y.H.; Liu, H.M.; Peir, J.J.; Jiang, S.H.; Chang, C.W.; Liu, C.S.; Lin, K.H.; Wang, S.J.; Chu, P.Y.; Lo, W.L.; Kao, S.Y.; Yen, S.H. Fractionated Boron Neutron Capture Therapy in Locally Recurrent Head and Neck Cancer: A Prospective Phase I/II Trial. Int J Radiat Oncol Biol Phys 2016, 95, 396-403. DOI: 10.1016/j.ijrobp.2016.02.028.
  14. Watanabe, T.; Hattori, Y.; Ohta, Y.; Ishimura, M.; Nakagawa, Y.; Sanada, Y.; Tanaka, H.; Fukutani, S.; Masunaga, S.I.; Hiraoka, M.; Suzuki, M.; Kirihata, M. Comparison of the pharmacokinetics between L-BPA and L-FBPA using the same administration dose and protocol: a validation study for the theranostic approach using [18F]-L-FBPA positron emission tomography in boron neutron capture therapy. BMC Cancer. 2016, 16, 859. DOI: 10.1186/s12885-016-2913-x.
  15. Moss, L. Critical review, with an optimistic outlook, on boron neutron capture therapy (BNCT). Appl Radiat Isotop 2014, 88, 2-11. DOI: 10.1016/j.apradiso.2013.
  16. Xuan, S.; Vicente M.G.H. Recent advances in boron delivery agents for boron neutron capture therapy (BNCT). In Boron‐Based Compounds: Potential and Emerging Applications in Medicine; Hey‐Hawkins, E., Viñas Teixidor, C., Eds.; John Wiley & Sons Ltd.: Oxford, UK, 2018; pp. 298-342.

Reviewer: Since the main subject of this manuscript is boronated carbohydrates current reviews and main papers on this subject should be cited:

  1. Kotora, Carboranyl-saccharide Derivatives: Syntheses and Biological Evaluation, In: Handbook of Boron Science with Applications in Organometallics, Catalysis, Materials and Medicine. Vol. 4. Boron in Medicine (Eds. N. S. Hosmane, R. Eagling), World Scientific: London, 2018, pp. 69-99.
  1. A.V. Orlova, L. O. Kononov, Synthesis of conjugates of polyhedral boron compounds with carbohydrates, Russ. Chem. Rev., 2009, 78, 629-648.
  1. Satapathy, B. P. Dash, C. S. Mahanta, B. R. Swain, B. B. Jena, N. S. Hosmane, Glycoconjugates of polyhedral boron clusters, J. Organomet. Chem., 2015, 798, 13-23.

Response: We have modified the following part of the Introduction and added the mentioned authors in the References.

“Recently, a saccharide-conjugated carborane compound (galactosyl carborane) [35], boronated carbohydrates [36,37,38,39], and carbohydrate derivatives of the closo-dodecaborate anion [40,41,42] were reported as specific targeting agents for potential application to BNCT.”

  1.  
  2.  
  3.  
  4.  
  5.  
  6.  
  7.  
  8.  
  9.  
  10.  
  11.  
  12.  
  13.  
  14.  
  15.  
  16.  
  17.  
  18.  
  19. Lai, C.H.; Lin. Y.C.; Chou, F.I.; Liang, C.F.; Lin, E.W.; Chuang, Y.J.; Lin, C.C. Design of multivalent galactosyl carborane as a targeting specific agent for potential application to boron neutron capture therapy. Chem Commun (Camb) 2012, 48, 612-614. DOI: 10.1039/c1cc14447b.
  20. Orlova, A.V.; Kononov, L.O. Synthesis of conjugates of polyhedral boron compounds with carbohydrates. Russ Chem Rev 2009, 78, 629-648.
  21. Marepally, R.; Yao, M.-L.; Kabalka, G.W. Boronated carbohydrate derivatives as potential boron neutron capture therapy reagents. Future Med Chem 2013, 5, 693-704. DOI: 10.4155/fmc.13.39.
  22. Satapathy, R.; Dash, B.P.; Mahanta, C.S.; Swain, B.R.; Jena, B.B.; Hosmane, N.S.; Glycoconjugates of polyhedral boron clusters. J Organomet Chem 2015, 798, 13-23. DOI: 10.1016/j.jorganchem.2015.06.027.
  23. Kotora, M. Carboranyl-saccharide Derivatives: Syntheses and Biological Evaluation. In Handbook of Boron Science with Applications in Organometallics, Catalysis, Materials and Medicine. Hosmane, N.S., Eagling, R., Eds.; World Scientific: London, UK, 2018; Volume 4 (Boron in Medicine), pp. 69-99.
  24. Lechtenberg, B.; Gabel, D. Synthesis of a (B12H11S)2− containing glucuronoside as potential prodrug for BNCT. J Organomet Chem 2005, 690, 2780-2782. DOI: 10.1016/j.jorganchem.2005.01.042.
  25. Orlova, A.V.; Kondakov, N.N.; Zinin, A.I.; Kimel', B.G.; Kononov, L.O.; Sivaev, I.B.; Bregadze, V.I. A uniform approach to the synthesis of carbohydrate conjugates of polyhedral boron compounds as potential agents for boron neutron capture therapy. Russ J Bioorg Chem 2006, 32, 568-577.
  26. Imperio, D.; Muz, B.; Azab, A.K.; Fallarini, S.; Lombardi, G.; Panza, L. A Short and Convenient Synthesis of closo‐Dodecaborate Sugar Conjugates. Eur J Org Chem 2019, 7228-7232. DOI: 10.1002/ejoc.202000042.

Reviewer 2 Report

The presented manuscript makes a useful contribution to the search for new compounds for BNCT. Both in vitro and in vivo biological experiment is well done, however the background of the study is not fully reflected in the introduction. The synthetic part also needs correction.

Surprising is the lack of references on foundational books and reviews on BNCT (See below) in the introduction except Ref. 4.

Neutron Capture Therapy. Principles and Applications (Eds. W. Sauerwein, A. Wittig, R. Moss, Y. Nakagawa), Springer-Verlag: Berlin, 2012.

  1. L. Moss, Critical review, with an optimistic outlook, on boron neutron capture therapy (BNCT), Appl. Radiat. Isotop.,201488, 2-11.
  2. F. Barth, Z. Zhang, T. Liu, A realistic appraisal of boron neutron capture therapy as a cancer treatment modality, Cancer Commun., 201838, 36.
  3. Xuan, M. G. H. Vicente, Recent advances in boron delivery agents for boron neutron capture therapy (BNCT), In: BoronBased Compounds: Potential and Emerging Applications in Medicine(Eds. E. Hey‐Hawkins, C. Viñas Teixidor), John Wiley & Sons Ltd.: Oxford, 2018, pp. 298-342.

The main reviews on boronated carbohydrates should be cited as well.

A.V. Orlova, L. O. Kononov, Synthesis of conjugates of polyhedral boron compounds with carbohydrates, Russ. Chem. Rev.200978, 629-648.

  1. R. Marepally, M.-L. Yao, G. W. Kabalka, Boronated carbohydrate derivatives as potential boron neutron capture therapy reagents, Future Med. Chem.20135, 693-704
  2. Satapathy, B. P. Dash, C. S. Mahanta, B. R. Swain, B. B. Jena, N. S. Hosmane, Glycoconjugates of polyhedral boron clusters, J. Organomet. Chem.2015798, 13-23.
  3. Kotora, Carboranyl-saccharide derivatives: Syntheses and biological evaluation, In: Handbook of Boron Science with Applications in Organometallics, Catalysis, Materials and Medicine. Vol. 4. Boron in Medicine(Eds. N. S. Hosmane, R. Eagling), World Scientific: London, 2018, pp. 69-99.

The papers on carbohydrate derivatives of the closo-dodecaborate anion should be mentioned as well

  1. Lechtenberg, D. Gabel, J. Organomet. Chem.2005690, 2780-2782.
  2. V. Orlova et al., Russ. J. Bioorg. Chem.200632, 568-577.
  3. Imperio et al., Eur. J. Org. Chem., 2019, 7228-7232.

The sentence starting on line 64 is not clear – «To this end, we have developed…» None of indicated here references are related either to the authors or to closo-dodecaborate. This leads to misunderstanding.

The commercial manufacturers of all the main commercial reagents used should be indicated in Section 2.1, as well as the literature methods for the synthesis of non-commercial reagents. In particular, the references on synthesis of compound 5 should be added. The given patent WO2017164334 is in Japan and therefore original references published in scientific journals should be added:

  1. V. Orlova et al., Appl. Organomet. Chem.200721, 98-100.
  2. A. Semioshkin et al., Collect. Czech. Chem. Commun.200772, 1717-1724.

The synthesis of compound 1 has been described several times in the literature [for example, see S. H. Khan et al., Carbohydrate Res.1990205, 385-397; S. Traboni et al., Beilstein J. Org. Chem.,201612, 2748-2756]. Therefore, it is enough to provide the reference on its synthesis.

Since Cells is not chemical journal, it is necessary to provide a transcript of all acronyms used (including in the schemes) in Section 2.1.

It is not clear why the mass spectral data for compound MMT1242 are verified in the positive mode, while for the analogous compound NBD1242 in the negative mode. This requires an explanation. In addition, the calculated masses of the compounds and their formulas should be given.

From the text of the article, it is unclear whether the authors of collection-10 worked with enriched compounds or with compounds with the natural content of this isotope.

General comments on the compound design.

With the design proposed by the authors, it is difficult to expect that compound MMT1242/NBD1242 will be able to accumulate in cancer cells via GLUT1-receptor mechanism due to over modification of the parent mannopyranoside.

In general, considering that the obtained compounds contain 36 times more boron atoms than BPA and provide only ~ 3 times better boron uptake at the same molar concentration, their design is very far from optimal. In this regard, the study of compounds with a lower degree of modification is of interest.

The increased retention of the compounds obtained is not surprising due to their highly negative charge. The similar effect was described earlier for various radiometal chelators [See K. Westerlund et al., Mol. Pharmaceutics, 201613, 1668-1678; S. S. Rinne et al., Int. J. Mol. Sci.202021, 1972].

Author Response

The presented manuscript makes a useful contribution to the search for new compounds for BNCT. Both in vitro and in vivo biological experiment is well done, however the background of the study is not fully reflected in the introduction. The synthetic part also needs correction.

Response:  We thank the reviewer for positive comments and helpful suggestions. We have taken all these comments and suggestions into account and they have improved our manuscript considerably.

Reviewer: Surprising is the lack of references on foundational books and reviews on BNCT (See below) in the introduction except Ref. 4.

Neutron Capture Therapy. Principles and Applications (Eds. W. Sauerwein, A. Wittig, R. Moss, Y. Nakagawa), Springer-Verlag: Berlin, 2012.

  1. L. Moss, Critical review, with an optimistic outlook, on boron neutron capture therapy (BNCT), Appl. Radiat. Isotop.,201488, 2-11.
  2. F. Barth, Z. Zhang, T. Liu, A realistic appraisal of boron neutron capture therapy as a cancer treatment modality, Cancer Commun., 201838, 36.
  3. Xuan, M. G. H. Vicente, Recent advances in boron delivery agents for boron neutron capture therapy (BNCT), In: BoronBased Compounds: Potential and Emerging Applications in Medicine(Eds. E. Hey‐Hawkins, C. Viñas Teixidor), John Wiley & Sons Ltd.: Oxford, 2018, pp. 298-342.

Response: We have modified the following part of the Introduction and added the mentioned authors in the References here and further in the text.

“Boron neutron capture therapy (BNCT) is a binary anticancer technology that was first proposed by Locher in 1936 and has since been further developed and improved by a number of research groups all over the world. It commands attention as a state-of-the-art, minimally invasive cancer treatment that functions as a single cell level-selective tumoricidal therapy [1,2,3,4,5,6,7]. It employs a stable boron-10 (10B) isotope-containing compound injected into the patient and neutron irradiation produced at a nuclear reactor (or accelerator) and, once the appropriate form of 10B-boron-containing compound has selectively accumulated in malignant tumor cells by a drug delivery system (DDS), the subsequent neutron irradiation of the tumor area induces alpha particles and Li nuclei emission within a 10 micrometer span, limiting the BNCT effect to tumor cells with accumulated boron and sparing the surrounding normal cells without boron accumulation [2,3,4,8,9]. This tumor-selective, high-dose irradiation may improve local control and survival after the initial treatment. Phase I/II clinical trials of BNCT were conducted for patients with glioblastoma and the early results suggest that BNCT is effective in selected cases [10,11,12].

Until recently, clinical studies using boronophenylalanine (BPA) and sodium borocaptate (BSH) as 10B delivery agents for the neutron capture reaction have been conducted [3,8,10,13]. However, continuous administration at high concentrations is needed to maintain sufficient 10B concentration within tumors [14]. For effective BNCT, ≥ 20 μg of 10B per gram of tumor must be selectively delivered to the tumor cells (~109 atoms per cell), reside intracellularly for a sufficient amount of time, and simultaneously clear from surrounding normal tissues to ideally attain a tumor-to-normal tissue ratio of 3-4 to 1 [2,3,4,5,15,16].”

  1. Locher G. Biological effects and therapeutic possibilities of neutrons. Am J Roentgenol Radium Ther 1936, 36, 1-13.
  2. Soloway, A.H.; Tjarks, W.; Barnum, B.A.; Rong, F.G.; Barth, R.F.; Codogni, I.M.; Wilson, J.G. The chemistry of neutron capture therapy. Chem Rev 1998, 98, 1515–1562. DOI: 10.1021/cr941195u.
  3. Sauerwein, W.A.G.; Wittig, A.; Moss, R.; Nakagawa, Y. Neutron Capture Therapy. Principles and Applications. Springer-Verlag: Berlin, Heidelberg, Germany. 2012.
  4. Sivaev, I.B.; Bregadze, V.I. Polyhedral boranes for medical applications: Current status and perspectives. Eur J Inorg Chem 2009, 11, 1433-1450. DOI: 10.1002/ejic.200900003.
  5. Barth, R.F.; Zhang, Z.; Liu, T. A realistic appraisal of boron neutron capture therapy as a cancer treatment modality. Cancer Commun (Lond) 2018, 38, 36. DOI: 10.1186/s40880-018-0280-5.
  6. Kreiner, A.J.; Bergueiro, J.; Cartelli, D.; Baldo, M.; Castell, W.; Asoia, J.G.; Padulo, J.; Suárez Sandín, J.C.; Igarzabal, M.; Erhardt, J.; Mercuri, D.; Valda, A.A.; Minsky, D.M.; Debray, M.E.; Somacal, H.R.; Capoulat, M.E.; Herrera, M.S.; Del Grosso, M.F.; Gagetti, L.; Anzorena, M.S.; Canepa, N.; Real, N.; Gun, M.; Tacca, H. Present status of Accelerator-Based BNCT. Rep Pract Oncol Radiother 2016, 21, 95-101. DOI: 10.1016/j.rpor.2014.11.004.
  7. Taskaev, S. Development of an accelerator-based epithermal neutron source for boron neutron capture therapy. Phys Part Nucl 2019, 50, 569–575. DOI: 10.1134/S1063779619050228.
  8. Barth, R.F.; Mi, P.; Yang, W. Boron delivery agents for neutron capture therapy of cancer. Cancer Commun(Lond) 2018, 38, 35. DOI: 10.1186/s40880-018-0299-7.
  9. Sato, E.; Zaboronok, A.; Yamamoto, T.; Nakai, K.; Taskaev, S.; Volkova, O.; Mechetina, L.; Taranin, A.; Kanygin, V.; Isobe, T.; Mathis, B. J.; & Matsumura, A. Radiobiological response of U251MG, CHO-K1 and V79 cell lines to accelerator-based boron neutron capture therapy. J Radiat Res 2018, 59, 101-107. DOI: 10.1093/jrr/rrx071.
  10. Yamamoto, T.; Nakai, K.; Tsurubuchi, T.; Matsuda, M.; Shirakawa, M.; Zaboronok, A.; Endo, K.; Matsumura, A. Boron neutron capture therapy for newly diagnosed glioblastoma: a pilot study in Tsukuba. Appl Radiat Isot 2009, 67(7-8 Suppl), S25-26. DOI: 10.1016/j.apradiso.2009.03.011.
  11. Kawabata, S.; Miyatake, S.; Hiramatsu, R.; Hirota, Y.; Miyata, S.; Takekita, Y.; Kuroiwa, T.; Kirihata, M.; Sakurai, Y.; Maruhashi, A.; Ono, K. Phase II clinical study of boron neutron capture therapy combined with X-ray radiotherapy/temozolomide in patients with newly diagnosed glioblastoma multiforme--study design and current status report. Appl Radiat Isot 2011, 69, 1796-1799. DOI: 10.1016/j.apradiso.2011.03.014.
  12. Miyatake, S.; Kawabata, S.; Hiramatsu, R.; Kuroiwa, T.; Suzuki, M.; Kondo, N.; Ono, K. Boron Neutron Capture Therapy for Malignant Brain Tumors. Neurol Med Chir (Tokyo) 2016, 56, 361-371. DOI: 10.2176/nmc.ra.2015-0297.
  13. Wang, L.W.; Chen, Y.W.; Ho, C.Y.; Hsueh Liu, Y.W.; Chou, F.I.; Liu, Y.H.; Liu, H.M.; Peir, J.J.; Jiang, S.H.; Chang, C.W.; Liu, C.S.; Lin, K.H.; Wang, S.J.; Chu, P.Y.; Lo, W.L.; Kao, S.Y.; Yen, S.H. Fractionated Boron Neutron Capture Therapy in Locally Recurrent Head and Neck Cancer: A Prospective Phase I/II Trial. Int J Radiat Oncol Biol Phys 2016, 95, 396-403. DOI: 10.1016/j.ijrobp.2016.02.028.
  14. Watanabe, T.; Hattori, Y.; Ohta, Y.; Ishimura, M.; Nakagawa, Y.; Sanada, Y.; Tanaka, H.; Fukutani, S.; Masunaga, S.I.; Hiraoka, M.; Suzuki, M.; Kirihata, M. Comparison of the pharmacokinetics between L-BPA and L-FBPA using the same administration dose and protocol: a validation study for the theranostic approach using [18F]-L-FBPA positron emission tomography in boron neutron capture therapy. BMC Cancer. 2016, 16, 859. DOI: 10.1186/s12885-016-2913-x.
  15. Moss, L. Critical review, with an optimistic outlook, on boron neutron capture therapy (BNCT). Appl Radiat Isotop 2014, 88, 2-11. DOI: 10.1016/j.apradiso.2013.
  16. Xuan, S.; Vicente M.G.H. Recent advances in boron delivery agents for boron neutron capture therapy (BNCT). In Boron‐Based Compounds: Potential and Emerging Applications in Medicine; Hey‐Hawkins, E., Viñas Teixidor, C., Eds.; John Wiley & Sons Ltd.: Oxford, UK, 2018; pp. 298-342.

Reviewer: The main reviews on boronated carbohydrates should be cited as well.

A.V. Orlova, L. O. Kononov, Synthesis of conjugates of polyhedral boron compounds with carbohydrates, Russ. Chem. Rev.200978, 629-648.

  1. R. Marepally, M.-L. Yao, G. W. Kabalka, Boronated carbohydrate derivatives as potential boron neutron capture therapy reagents, Future Med. Chem.20135, 693-704
  2. Satapathy, B. P. Dash, C. S. Mahanta, B. R. Swain, B. B. Jena, N. S. Hosmane, Glycoconjugates of polyhedral boron clusters, J. Organomet. Chem.2015798, 13-23.
  3. Kotora, Carboranyl-saccharide derivatives: Syntheses and biological evaluation, In: Handbook of Boron Science with Applications in Organometallics, Catalysis, Materials and Medicine. Vol. 4. Boron in Medicine(Eds. N. S. Hosmane, R. Eagling), World Scientific: London, 2018, pp. 69-99.

The papers on carbohydrate derivatives of the closo-dodecaborate anion should be mentioned as well

  1. Lechtenberg, D. Gabel, J. Organomet. Chem.2005690, 2780-2782.
  2. V. Orlova et al., Russ. J. Bioorg. Chem.200632, 568-577.
  3. Imperio et al., Eur. J. Org. Chem., 2019, 7228-7232.

Response: We have modified the following part of the Introduction and added the mentioned authors in the References.

“Recently, a saccharide-conjugated carborane compound (galactosyl carborane) [35], boronated carbohydrates [36,37,38,39], and carbohydrate derivatives of the closo-dodecaborate anion [40,41,42] were reported as specific targeting agents for potential application to BNCT.”

  1.  
  2.  
  3.  
  4.  
  5.  
  6.  
  7.  
  8.  
  9.  
  10.  
  11.  
  12.  
  13.  
  14.  
  15.  
  16.  
  17.  
  18.  
  19.  
  20.  
  21.  
  22.  
  23.  
  24.  
  25.  
  26.  
  27.  
  28.  
  29.  
  30.  
  31.  
  32.  
  33.  
  34.  
  35.  
  36.  
  37. Lai, C.H.; Lin. Y.C.; Chou, F.I.; Liang, C.F.; Lin, E.W.; Chuang, Y.J.; Lin, C.C. Design of multivalent galactosyl carborane as a targeting specific agent for potential application to boron neutron capture therapy. Chem Commun (Camb) 2012, 48, 612-614. DOI: 10.1039/c1cc14447b.
  38. Orlova, A.V.; Kononov, L.O. Synthesis of conjugates of polyhedral boron compounds with carbohydrates. Russ Chem Rev 2009, 78, 629-648.
  39. Marepally, R.; Yao, M.-L.; Kabalka, G.W. Boronated carbohydrate derivatives as potential boron neutron capture therapy reagents. Future Med Chem 2013, 5, 693-704. DOI: 10.4155/fmc.13.39.
  40. Satapathy, R.; Dash, B.P.; Mahanta, C.S.; Swain, B.R.; Jena, B.B.; Hosmane, N.S.; Glycoconjugates of polyhedral boron clusters. J Organomet Chem 2015, 798, 13-23. DOI: 10.1016/j.jorganchem.2015.06.027.
  41. Kotora, M. Carboranyl-saccharide Derivatives: Syntheses and Biological Evaluation. In Handbook of Boron Science with Applications in Organometallics, Catalysis, Materials and Medicine. Hosmane, N.S., Eagling, R., Eds.; World Scientific: London, UK, 2018; Volume 4 (Boron in Medicine), pp. 69-99.
  42. Lechtenberg, B.; Gabel, D. Synthesis of a (B12H11S)2− containing glucuronoside as potential prodrug for BNCT. J Organomet Chem 2005, 690, 2780-2782. DOI: 10.1016/j.jorganchem.2005.01.042.
  43. Orlova, A.V.; Kondakov, N.N.; Zinin, A.I.; Kimel', B.G.; Kononov, L.O.; Sivaev, I.B.; Bregadze, V.I. A uniform approach to the synthesis of carbohydrate conjugates of polyhedral boron compounds as potential agents for boron neutron capture therapy. Russ J Bioorg Chem 2006, 32, 568-577.
  44. Imperio, D.; Muz, B.; Azab, A.K.; Fallarini, S.; Lombardi, G.; Panza, L. A Short and Convenient Synthesis of closo‐Dodecaborate Sugar Conjugates. Eur J Org Chem 2019, 7228-7232. DOI: 10.1002/ejoc.202000042.

Reviewer: The sentence starting on line 64 is not clear – «To this end, we have developed…» None of indicated here references are related either to the authors or to closo-dodecaborate. This leads to misunderstanding.

Response: We have modified that sentence in the Introduction as follows.

“We have developed novel boron-containing low molecular compounds efficient for accumulation and retention in tumors, one of which is a boron-containing α-D-mannopyranoside derivative with three closo-dodecarborates each carrying 12 boron atoms [17,18].”

  1. Tietze, L.F.; Griesbach, U.; Bothe, U.; Nakamura, H., Yamamoto, Y. Novel carboranes with a DNA binding unit for the treatment of cancer by boron neutron capture therapy. Chembiochem 2002; 3(2-3), 219-225. DOI: 10.1002/1439-7633(20020301)3:2/3%3C219::AID-CBIC219%3E3.0.CO;2-%23.
  2. Calabrese, G.; Daou, A.; Barbu, E.; Tsibouklis, J. Towards carborane-functionalised structures for the treatment of brain cancer. Drug Discov Today 2018, 23, 63-75. DOI: 10.1016/j.drudis.2017.08.009.

Reviewer: The commercial manufacturers of all the main commercial reagents used should be indicated in Section 2.1, as well as the literature methods for the synthesis of non-commercial reagents. In particular, the references on synthesis of compound 5 should be added. The given patent WO2017164334 is in Japan and therefore original references published in scientific journals should be added:

  1. V. Orlova et al., Appl. Organomet. Chem.200721, 98-100.
  2. A. Semioshkin et al., Collect. Czech. Chem. Commun.200772, 1717-1724.

Response: We have added the information to Materials and Methods and modified as follows.

2.1. Chemical synthesis

For chemical synthesis, cesium dodecaborate and 10B-enriched cesium dodecaborate were purchased from Katchem Ltd. (Prague, Czech Republic), QuadraPure® IDA was obtained from Sigma-Aldrich Inc. (St. Louis, MO, USA), Amberlite® IR120B Na was obtained from Organo Corp. (Tokyo, Japan), all other reagents, including methyl α-D-mannopyranoside and 4-fluoro-7-nitro-2,1,3-benzoxadiazole, were purchased from Tokyo Chemical Industry Co., Ltd. (Tokyo, Japan), and all the solvents were purchased from Kanto Chemical Co., Inc. (Tokyo, Japan).”

“The closo-dodecaborate unit (5) that was previously synthesized by Orlova et al. (2007) and Semioshkin et al. (2007) was here prepared according to our methods described in the patent (WO2017164334) and, according to our modifications, contains a cesium salt to provide sufficient water solubility in contrast to the previously reported compound that contained a tetrabutylammonium salt [43,44].”

  1. Orlova, A.V.; Kondakov, N.N.; Kimel’, B.G.; Kononov, L.O.; Kononova, E.G.; Sivaev, I.B.; Bregadze, V.I. Synthesis of novel derivatives of closo-dodecaborate anion with azido group at the terminal position of the spacer. Appl Organomet Chem 2007, 21, №2, 98-100. DOI: 10.1002/aoc.1151.
  2. Semioshkin, A.A.; Osipov, S.N.; Grebenyuk, J.N.; Nizhnik, E.A.; Godovikov, I.A.; Shchetnikov, G.T.; Bregadze, V.I. An Effective Approach to 1,2,3-Triazole-Containing 12-Vertex closo-Dodecaborates. Collect Czech Chem Commun 2007, 72, 1717-1724. DOI: 10.1135/cccc20071717.

Reviewer: The synthesis of compound 1 has been described several times in the literature [for example, see S. H. Khan et al., Carbohydrate Res.1990205, 385-397; S. Traboni et al., Beilstein J. Org. Chem.,201612, 2748-2756]. Therefore, it is enough to provide the reference on its synthesis.

Response: We have added the mentioned references to the synthesis of compound 1 in Materials and Methods as follows.

Methyl 6-O-(tert-butyldiphenylsilanyl)-α-D-mannopyranoside (1) [45,46]

  1. Khan, S.H.; Abbas, S.A.; Matta, K.L. Synthesis of 4-nitrophenyl O-(2-acetamido-2-deoxy-beta-D-glucopyranosyl)- (1----2)-O-(4-O-methyl-alpha-D-mannopyranosyl)-(1----6)-beta-D-glucopyr Anoside. A Potential Specific Acceptor-Substrate for N-acetylglucosaminyltransferase-V (GnT V). Carbohydrate Res 1990, 205, 385-397. DOI: 10.1016/0008-6215(93)84079-l.
  2. Traboni, S.; Bedini, E.; Iadonisi, A. Orthogonal protection of saccharide polyols through solvent-free one-pot sequences based on regioselective silylations. Beilstein J Org Chem 2016, 12, 2748-2756. DOI: 10.3762/bjoc.12.271.

Reviewer: Since Cells is not chemical journal, it is necessary to provide a transcript of all acronyms used (including in the schemes) in Section 2.1.

Response: We have added the transcripts of all the acronyms used in Section 2.1. as follows.

Materials and Methods

“Figure 1. Synthesis of MMT1242. Reaction conditions and yields: a) tert-butylchlorodiphenylsilane, imidazole, N,N-dimethylformamide, 5˚C to rt (quant.); b) sodium hydride, N,N-dimethylformamide, 5˚C to rt, then propargyl bromide, cat. tetrabutylammonium iodide, 5˚C to rt (68%); c) tetrabutylammonium fluoride, tetrahydrofuran, 5˚C to rt (91%); d) sodium hydride, N,N-dimethylformamide, 5˚C to rt, then benzyl bromide, rt (53%); e) 5, copper (II) sulfate pentahydrate, sodium L-ascorbate, tert-butanol/ H2O (1/2, v/v), 70˚C; f) Amberlite® IR120B Na, H2O, (25%, 2 steps).

2.1.1. Synthesis of MMT1242

Methyl 6-O-(tert-butyldiphenylsilanyl)-α-D-mannopyranoside (1) [45,46]

To a solution of methyl α-D-mannopyranoside (5.00 g, 25.8 mmol) in N,N-dimethylformamide (5 mL) was added imidazole (2.11 g, 31.0 mmol) at room temperature. After tert-butylchlorodiphenylsilane (7.81 g, 28.4 mmol) was added at 0˚C (ice-water bath), the mixture was stirred at room temperature overnight. The reaction mixture was diluted with diethyl ether and washed with water, followed by 10% citric acid and 15% brine. Then, the organic layer was dried over anhydrous magnesium sulfate. Afterfiltration, the filtrate was concentrated under reduced pressure. The obtained residue was purified by silica gel column chromatography using ethyl acetate/methanol to give compound 1 (12.5 g, 29.0 mmol, quantitative yield).

………..

Methyl 6-O-(tert-butyldiphenylsilanyl)-2,3,4-tris-O-(prop-2-ynyl)- α-D-mannopyranoside (2)

To a mixture of sodium hydride (1.00 g, 60% suspension in oil, 25.0 mmol) and dehydrated N,N-dimethylformamide (1.5 mL), a solution of compound 1 (3.00 g, 6.94 mmol) in N,N-dimethylformamide (9 mL) was added at 0˚C (ice-water bath). After stirring of the mixture at room temperature for 20 minutes, propargyl bromide (2.35 mL, 31.2 mmol) and tetrabutylammonium iodide (933 mg, 2.5 mmol) were added at 0˚C (ice-water bath), and the mixture was stirred at room temperature overnight. The reaction mixture was diluted with diethyl ether and washed with water, followed by 10% citric acid and 15% brine. Then, the organic layer was dried over anhydrous magnesium sulfate. After filtration, the filtrate was concentrated under reduced pressure. The obtained residue was purified by silica gel column chromatography using hexane/ethyl acetate to give compound 2 (2.60 g, 4.75 mmol, 68%).

…….

Figure 2. Synthesis of NBD1242. Reaction conditions and yields: a) phthalimide, triphenylphosphine, diethyl azodicarboxylate (40% in toluene), tetrahydrofuran, 5˚C to rt; b) hydrazine monohydrate, ethanol, rt, then preparative RP-HPLC (containing 0.05% trifluoroacetic acid), (50%); c) 4-fluoro-7-nitro-2,1,3-benzoxadiazole, triethylamine, N,N-dimethylformamide, 50˚C (38%); d) 5, copper (II) sulfate pentahydrate, sodium L-ascorbate, tert-butanol/ H2O (1/2, v/v), 70˚C; e) H2O, Amberlite® IR120B Na (19%, 2 steps).

Methyl 2,3,4-tris-O-(prop-2-ynyl)- α-D-mannopyranoside (3)

To a solution of compound 2 (2.58 g, 4.72 mmol) in tetrahydrofuran (24 mL) was added 1M tetrabutylammonium fluoride in tetrahydrofuran (9.44 mL, 9.44 mmol) at 0˚C (ice-water bath). After stirring of the mixture at room temperature for 6 hours, the reaction mixture was diluted with dichloromethane, and washed with water followed by 15% brine. Then, the organic layer was dried over anhydrous magnesium sulfate. After filtration, the filtrate was concentrated under reduced pressure. The obtained residue was purified by silica gel column chromatography using hexane/ethyl acetate to give compound 3 (1.32 g, 4.29 mmol, 91%).

………

Methyl 6-O-benzyl-2,3,4-tris-O-(prop-2-ynyl)- α-D-mannopyranoside (4)

To a solution of compound 3 (150 mg, 0.486 mmol) in N,N-dimethylformamide (2.4 mL) was added sodium hydride (35.0 mg, 60% suspension in oil, 0.583 mmol) at 0˚C (ice-water bath), and the mixture was stirred at room temperature for 30 minutes. …………..

To a solution of compound 3 (426 mg, 1.38 mmol) in tetrahydrofuran (14 mL) was added triphenylphosphine (543 mg, 2.07 mmol), diethyl azodicarboxylate (0.940 mL, 40% in toluene, 2.07 mmol) and phthalimide (305 mg, 2.07 mmol) at 0˚C (ice-water bath). After stirring of the mixture at room temperature for 40 minutes, triphenylphosphine (181 mg, 0.690 mmol), diethyl azodicarboxylate (0.627 mL, 40% in toluene, 1.38 mmol) and phthalimide (102 mg, 0.690 mmol) were added at 0˚C (ice-water bath). The mixture was stirred at room temperature for 25 minutes then concentrated under reduced pressure. The obtained residue was purified by silica gel column chromatography using hexane/ ethyl acetate to give compound 6 (743 mg, 1.70 mmol) as crude material.

……….

To a solution of compound 6 (604 mg, 1.38 mmol) in ethanol (7 mL) was added hydrazine monohydrate (88.5 mg, 2.76 mmol). The mixture was stirred at room temperature overnight. After filtration, the filtrate was concentrated under reduced pressure. The obtained residue was purified by reverse-phase HPLC (water/acetonitrile, containing 0.05% trifluoroacetic acid) to give a mixture of compound 7. The mixture was diluted with ethyl acetate and extracted with 0.2N HCl. After concentration of the aqueous layer under reduced pressure, the obtained residue was purified by reverse phase HPLC (water/acetonitrile, containing 0.05% trifluoroacetic acid) to give compound 7 (211 mg, 0.686 mmol, 50%).

…….

To a solution of compound 7 (96.4 mg, 0.314 mmol) in N,N-dimethylformamide (1 mL) was added triethylamine (127 mg, 1.26 mmol) and 4-fluoro-7-nitro-2,1,3-benzoxadiazole (86.1 mg, 0.470 mmol). ”

Reviewer: It is not clear why the mass spectral data for compound MMT1242 are verified in the positive mode, while for the analogous compound NBD1242 in the negative mode. This requires an explanation. In addition, the calculated masses of the compounds and their formulas should be given.

Response: The previous mass spectral ( MS ) data showed only the highest scores of MMT1242 in the positive mode, and those of NBD1242 in the negative mode. We have corrected the expressions of MS data of MMT1242 and NBD1242 and added the calculated masses of the compounds and their formulas in the Materials and Methods as follows.

For MMT1242,

“MS (ESI, m/z): 651.0 [M+6Na-2Na]2-; 1372 [M+7Na]+. C35H83B36N9Na6O12. Calculated: m/z 651.5 [M+6Na-2Na]2-; 1372 [M+7Na]+.”

For NBD1242,

“MS (ESI, m/z): 687.9 [M+6Na-2Na]2-; 1397 [M+6Na-Na]-. C34H79B36N13Na5O14. Calculated: m/z 687.5 [M+6Na-2Na]2-; 1398 [M+6Na-Na]-.”

Reviewer: From the text of the article, it is unclear whether the authors of collection-10 worked with enriched compounds or with compounds with the natural content of this isotope.

Response: We have added the following explanation in Materials and Methods.

“MMT1242 containing 10B isotope was used in fluorescence imaging of 10B distribution in vivo and in irradiation experiments while, in all other experiments, MMT1242 containing natural boron was used.”

Reviewer: General comments on the compound design.

With the design proposed by the authors, it is difficult to expect that compound MMT1242/NBD1242 will be able to accumulate in cancer cells via GLUT1-receptor mechanism due to over modification of the parent mannopyranoside.

Response: We thank the Reviewer for the valuable comment. We have done the study on intracellular boron uptake in CT26 tumor cells with GLUT1 inhibition, and added the results in the manuscript. However, our results indicated that MMT1242 uptake may be GLUT1 independent as inhibition of GLUT1 receptor by cytochalasin B or downregulation of GLUT1 receptor by high glucose did not affect the uptake of MMT1242. Other possible mechanisms, such as endocytosis by mannose receptors on the plasma membrane of CT 26 mice colon carcinoma cells, may explain this result. Mannose receptors cannot be blocked by cytochalasin B.

Mannose receptors (MRs) are ubiquitously expressed on various types of cells [22]. CD206 (Cluster of Differentiation 206) expression in mouse colon was previously observed [48]. Mannose receptors are also expressed in CT26 colon tumor cells [31]. In spite of significant long-tailed mannopyranoside modifications, including N-(4-nitrobenzo-2-oxa-1,3-diazole) (NBD), these compounds were also selectively uptaken by mannose receptors [30]. Further uptake mechanisms should be elucidated by future studies.

  1. Sharma, V.; Ichikawa, M.; Freeze, H.H. Mannose metabolism: more than meets the eye. Biochem Biophys Res Commun 2014, 453, 220-228. DOI: 10.1016/j.bbrc.2014.06.021.
  2. Heinsbroek, S.E.; Squadrito, M.L.; Schilderink, R.; Hilbers, F.W.; Verseijden, C.; Hofmann, M.; Helmke, A.; Boon, L.; Wildenberg, M.E.; Roelofs, J.J.; Ponsioen, C.Y.; Peters, C.P.; Te Velde, A.A.; Gordon, S.; De Palma, M.; de Jonge, W.J. miR-511-3p, embedded in the macrophage mannose receptor gene, contributes to intestinal inflammation. Mucosal Immunol 2016, 9, 960-973. DOI: 10.1038/mi.2015.113.
  3. Xiong, M.; Lei, Q.; You, X.; Gao, T.; Song, X.; Xia, Y.; Ye, T.; Zhang, L.; Wang, N.; Yu, L. Mannosylated liposomes improve therapeutic effects of paclitaxel in colon cancer models. J Microencapsul 2017, 34, 513-521. DOI: 10.1080/02652048.2017.1339739.
  4. Jeong, H.S.; Na, K.S.; Hwang, H.; Oh, P.S.; Kim, D.H.; Lim, S.T.; Sohn, M.H.; Jeong, H.J. Effect of space length of mannose ligand on uptake of mannosylated liposome in RAW 264.7 cells: In vitro and in vivo studies. J Biomed Mater Res A 2014, 102, 4545-4553. DOI: 10.1002/jbm.a.35112.

Reviewer: In general, considering that the obtained compounds contain 36 times more boron atoms than BPA and provide only ~ 3 times better boron uptake at the same molar concentration, their design is very far from optimal. In this regard, the study of compounds with a lower degree of modification is of interest.

Response: We thank the Reviewer for the valuable comment. We have also discussed the possibility of further application of the compound in the manuscript (Discussion):

“Although we showed MMT1242 to have a broad and retentive distribution in diverse cancer cell lines and tissues, more precise studies on uptake and retention mechanisms in the tumor cells need to be evaluated before stepping up to human clinical trials.”

Reviewer: The increased retention of the compounds obtained is not surprising due to their highly negative charge. The similar effect was described earlier for various radiometal chelators [See K. Westerlund et al., Mol. Pharmaceutics, 201613, 1668-1678; S. S. Rinne et al., Int. J. Mol. Sci.202021, 1972].

Response: We thank the Reviewer for the valuable comment. We have added the following sentence in the Discussion.

“Another reason why MMT1242 showed increased retention in tumor cells might be its highly negative charge. Some compounds showed a similar effect in previous studies [57,58].”

  1.  
  2.  
  3.  
  4.  
  5.  
  6.  
  7.  
  8.  
  9.  
  10.  
  11.  
  12.  
  13.  
  14.  
  15.  
  16.  
  17.  
  18.  
  19.  
  20.  
  21.  
  22.  
  23.  
  24.  
  25.  
  26.  
  27.  
  28.  
  29.  
  30.  
  31. Westerlund, K.; Honarvar, H.; Norrström, E.; Strand, J.; Mitran, B.; Orlova, A.; Karlström, A.E.; Tolmachev, V. Increasing the Net Negative Charge by Replacement of DOTA Chelator with DOTAGA Improves the Biodistribution of Radiolabeled Second-Generation Synthetic Affibody Molecules. Mol Pharmaceutics 2016, 13, 1668-1678. DOI: 10.1021/acs.molpharmaceut.6b00089.
  32. Rinne, S.S.; Leitao, C.D.; Saleh-nihad, Z.; Mitran, B.; Tolmachev, V.; Ståhl, S.; Löfblom, J.; Orlova, A. Benefit of Later-Time-Point PET Imaging of HER3 Expression Using Optimized Radiocobalt-Labeled Affibody Molecules. In J Mol Sci 2020, 21, 1972. DOI: 10.3390/ijms21061972.

Reviewer 3 Report

Evaluation of A Novel Boron-Containing α-D mannopyranoside for BNCT

In this contribution, authors synthesize a novel boron-containing α-D-mannopyranoside, MMT1242, for BNCT and a fluorescent analogue NBD1242. They evaluated the uptake, intracellular distribution and retention of MMT1242 in culture cells, and analyzed biodistribution, tumor-to-normal tissue ratio, toxicity and effectiveness of BNCT in vivo. The authors to conclude that 10B-MMT1242 is a potential candidate for further investigation in clinical BNCT studies.

Publish after major revision

  • Authors claim compounds MMT1242 and NBD1242 as novel. However, the complete characterization is missing, only 1H-NMR data is given (no assignments were done) and ESI-MS spectra. Authors must include 11B-NMR data and also 13C NMR data, as well as mp and IR spectra.
  • Biodistribution and irradiation experiments were compared with BPA and BHD compounds, but the amounts of 10B in mg/kg administrated were not equivalent. Always the quantity of BPA was one half MMT1242. This condition should be included in the discussion (for example in 3.3.3 section, in the figure 9) and taken into account in the conclusions.
  • The authors claim that the reported compounds are low molecular weight but MS values of 1373 and 1367 are far from the corresponding values of BPA and BSH. The authors should put into context the use of the term “low molecular weight”.
  • The conclusions section should be shortened, authors include statements as “we can also propose another possibility that MMT1242 recognized with Mannose receptor might be C-H-π C-H-π stacking exists C-H groups interact with aromatic residues in the binding pocket by van der Waals forces between hydrogen bond with the receptor’s amino acids and the protein hydration shell” but none theoretical calculations were performed or reference cited to support this proposal.
  • There are several mistypes, for example but not all included: lines 232and 247 the exponent should be superindexed; line 373 10B instead of 10B; 258 he character “*” is used to indicate product whereas in other parts of the text “x” is used instead.

Author Response

In this contribution, authors synthesize a novel boron-containing α-D-mannopyranoside, MMT1242, for BNCT and a fluorescent analogue NBD1242. They evaluated the uptake, intracellular distribution and retention of MMT1242 in culture cells, and analyzed biodistribution, tumor-to-normal tissue ratio, toxicity and effectiveness of BNCT in vivo. The authors to conclude that 10B-MMT1242 is a potential candidate for further investigation in clinical BNCT studies.

Response:  We thank the reviewer for positive comments and helpful suggestions. We have taken all these comments and suggestions into account and they have improved our manuscript considerably.

Reviewer: Authors claim compounds MMT1242 and NBD1242 as novel. However, the complete characterization is missing, only 1H-NMR data is given (no assignments were done) and ESI-MS spectra. Authors must include 11B-NMR data and also 13C NMR data, as well as mp and IR spectra.

Response:  We have added 1H-NMR data in the Materials and Methods section as follows. However, we do not have 11B-NMR data and 13C-NMR data, as well as mp and IR spectra. In the current situation, it is difficult to access the equipment and therefore to perform additional measurements.

Materials and Methods

For MMT1242,

“MS (ESI, m/z): 651.0 [M+6Na-2Na]2-; 1372 [M+7Na]+. C35H83B36N9Na6O12. Calculated: m/z 651.5 [M+6Na-2Na]2-; 1372 [M+7Na]+.

1H NMR (400 MHz, deuterium oxide) δ 8.11 (s, 1H, triazole -CH), 8.07 (s, 1H, triazole -CH), 7.68 (s, 1H, triazole -CH), 7.33 – 7.24 (m, 5H, phenyl -CH), 4.79 – 4.73 (m, 4H), 4.63 – 4.56 (m, 1H, Man -CH), 4.56 – 4.49 (m, 5H), 4.44 – 4.30 (m, 4H), 3.94 – 3.73 (m, 8H), 3.68 – 3.38 (m, 17H, -OCH2-), 3.32 (s, 3H, -OCH3), 1.88 – 0.34 (m, 33H, -B12H11).”

For NBD1242,

“N-[[(3R,4S,6S)-6-methoxy-3,4,5-tris-O-[[1-[2-[2-(undecahydro-closo-dodecaboranyloxy)ethoxy]ethyl]triazol-4-yl]methyl]-tetrahydropyran-2-yl]methyl]-4-nitro-2,1,3-benzoxadiazol-7-amine hexasodium salt (NBD1242)

An operation similar to that of MMT1242 was performed using compound 8 (56.3 mg, 0.120 mmol) as a starting material to give NBD1242 (38.0 mg, 0.0273 mmol, 19%).

MS (ESI, m/z): 687.9 [M+6Na-2Na]2-; 1397 [M+6Na-Na]-. C34H79B36N13Na5O14. Calculated: m/z 687.5 [M+6Na-2Na]2-; 1398 [M+6Na-Na]-.

1H NMR (400 MHz, deuterium oxide) δ 8.48 (d, J = 9.0 Hz, 1H, NBD -CH), 8.17 (s, 1H, triazole -CH), 8.14 (s, 1H, triazole -CH), 8.05 (s, 1H, triazole -CH), 6.29 (d, J = 9.1 Hz, 1H, NBD -CH), 4.85 – 4.76 (m, 4H), 4.67 – 4.60 (m, 2H), 4.60 – 4.53 (m, 4H), 4.43 – 4.37 (m, 2H), 3.96 (t, J = 2.6 Hz, 1H, Man -CH), 3.91 – 3.73 (m, 8H), 3.72 – 3.65 (m, 1H, Man -CH), 3.65 – 3.30 (m, 15H, -OCH2-), 3.23 (s, 3H, -OCH3), 1.81 – 0.26 (m, 33H, -B12H11).”

Reviewer: Biodistribution and irradiation experiments were compared with BPA and BHD compounds, but the amounts of 10B in mg/kg administrated were not equivalent. Always the quantity of BPA was one half MMT1242. This condition should be included in the discussion (for example in 3.3.3 section, in the figure 9) and taken into account in the conclusions.

The authors claim that the reported compounds are low molecular weight but MS values of 1373 and 1367 are far from the corresponding values of BPA and BSH. The authors should put into context the use of the term “low molecular weight”.

The conclusions section should be shortened, authors include statements as “we can also propose another possibility that MMT1242 recognized with Mannose receptor might be C-H-π C-H-π stacking exists C-H groups interact with aromatic residues in the binding pocket by van der Waals forces between hydrogen bond with the receptor’s amino acids and the protein hydration shell” but none theoretical calculations were performed or reference cited to support this proposal.

Response: We have added the following paragraph in the Discussion.

“There are limitations to this study. First, the cutoff for a low-molecular-weight-boron compound is usually defined as below 104 [64]. Therefore, the MMT1242 molecular weight is low but, as it is higher than that of BPA, we have to consider the possibility of different biodistribution and tumor uptake dynamics among MMT1242, BPA and BSH due to their different molecular weights in future studies. Second, although the amounts of natural B and 10B in mg/kg equivalents administered in the form of BPA or MMT1242 in the biodistribution study were not the same, we chose only those doses of MMT1242 that were non-toxic and showed both good retention in tumor cells and adequate tumor-to-normal tissue accumulation ratio. According to the results of the biodistribution study, we did irradiation experiments using a low and a high dose of MMT1242 compared to BPA. Although the amounts of administered 10B were not equivalent in mg/kg to the high dose of MMT1242 and BPA in the irradiation study, the high dose of MMT1242 showed low toxicity and significant tumor inhibiting effect upon injection 24 hours before irradiation compared to the low doses of MMT1242 and BPA. Third, for human application of MMT1242 in BNCT, more precise toxicity studies based on animal experiments are needed. Although we showed MMT1242 to have a broad and retentive distribution in diverse cancer cell lines and tissues, more precise studies on uptake and retention mechanisms in the tumor cells need to be evaluated before stepping up to human clinical trials.”

  1. Carlsson, J.; Kullberg, E.B.; Capala, J.; Sjöberg, S.; Edwards, K.; Gedda, L. Ligand Liposomes and Boron Neutron Capture Therapy. J Neurooncol 2003;62(1-2):47-59. DOI: 10.1007/BF02699933.

Reviewer: There are several mistypes, for example but not all included: lines 232and 247 the exponent should be superindexed; line 373 10B instead of 10B; 258 he character “*” is used to indicate product whereas in other parts of the text “x” is used instead.

Response: We have corrected the mistypes as follows.

Page #6 line232;

“1.5x104 cells”, instead of “1.5x104 cells”

Page #7 line247;

“1x106 cells”, instead of “1x106 cells”

Page #7 line258;

“x”, instead of “*”

Page #11 line373;

10B”, instead of “10B”

Round 2

Reviewer 2 Report

The authors did a good job, which led to a significant improvement in the manuscript. I have only one remark left concerning the sentence on lines 68-70 - studies [17] and [18] were not performed by the authors of the manuscript and therefore, in order to avoid misunderstanding, these references should find another place in the text that corresponds to them.

Author Response

The authors did a good job, which led to a significant improvement in the manuscript. I have only one remark left concerning the sentence on lines 68-70 - studies [17] and [18] were not performed by the authors of the manuscript and therefore, in order to avoid misunderstanding, these references should find another place in the text that corresponds to them.

Response:  We thank the reviewer for positive comments and helpful suggestions. We have added the following sentence to the Introduction.

“To fulfill these criteria, various functionalized carborane derivatives have been recently developed, based on vision of carboranes as molecular moieties of choice for the synthesis of boron-delivery agents for BNCT due to their catabolic stability, relatively low toxicity, and a high boron content compared to BPA [17,18]”

Reviewer 3 Report

The authors should modify the abstract, concerning to clinical trials and accordingly to the included text .

Author Response

The authors should modify the abstract, concerning to clinical trials and accordingly to the included text.

Response:  We thank the reviewer for positive comments and helpful suggestions. We have modified the Abstract as follows.

“Boron neutron capture therapy (BNCT) is a unique anticancer technology that has demonstrated its efficacy in numerous phase I/II clinical trials with boronophenylalanine (BPA) and sodium borocaptate (BSH) used as 10B delivery agents. However, continuous drug administration at high concentrations is needed to maintain sufficient 10B concentration within tumors. To address the issue of 10B accumulation and retention in tumor tissue we developed MMT1242, a novel boron-containing α-D-mannopyranoside. We evaluated the uptake, intracellular distribution, and retention of MMT1242 in cultured cells and analyzed biodistribution, tumor-to-normal tissue ratio and toxicity in vivo. Fluorescence imaging using nitrobenzoxadiazole (NBD)-labeled MMT1242 and inductively coupled mass spectrometry (ICP-MS) were performed. The effectiveness of BNCT using MMT1242 was assessed in animal irradiation studies at the Kyoto University Research Reactor. MMT1242 showed a high uptake and broad intracellular distribution in vitro, longer tumor retention compared to BSH and BPA, and adequate tumor-to-normal tissue accumulation ratio and low toxicity in vivo. A neutron irradiation study with MMT1242 in a subcutaneous murine tumor model revealed a significant tumor inhibiting effect if injected 24 hours before irradiation. We therefore report that 10B-MMT1242 is a candidate for further clinical BNCT studies.”